# The evolution of stable silicon isotopes in a coastal carbonate aquifer, Rottnest Island, Western Australia

Ashley N. Martin[1,2,3,4,*], Karina Meredith[1,2,], Andy Baker[1,3] Marc D. Norman[5] and Eliza Bryan[1,2,3]

[1]Connected Waters Initiative Research Centre, UNSW Sydney, Sydney, NSW 2052, Australia.

[2]Australian Nuclear Science and Technology Organisation, Lucas Heights, NSW 2234, Australia.

[3]School of Biological, Earth and Environmental Sciences, UNSW Sydney, Sydney, NSW 2052, Australia.

[4]Institut für Mineralogie, Leibniz Universität Hannover, Callinstraße 3, 30167 Hannover, Germany.

[5]Research School of Earth Sciences, Australian National University, Canberra, ACT 2601, Australia.

*Correspondence to: a.martin@mineralogie.uni-hannover.de

**Abstract**. Dissolved silicon (dSi) is a key nutrient in the oceans, but there are few data available regarding Si isotopes in coastal aquifers. Here we investigate the Si isotopic composition of 12 fresh and 16 saline groundwater samples from Rottnest Island, Western Australia, which forms part of the world's most extensive aeolianite deposit (the Tamala Limestone Formation). Two bedrock samples were also collected from Rottnest Island for Si isotope analysis. The $\delta^{30}$Si values of groundwaters ranged from -0.4 to +3.6 ‰ with an (average: +1.6 ‰) and the rock samples were -0.8 and -0.1 ‰. The increase in $\delta^{30}$Si values in fresh groundwaters is attributed to the removal of the lighter Si isotopes into secondary minerals and potentially also adsorption onto Fe (oxy)hydroxides. The positive correlations between $\delta^{30}$Si values and dSi concentrations ($\rho = 0.59$, p = 0.02) and $\delta^{30}$Si values and Cl, but not dSi and Cl concentrations, are consistent with vertical mixing between the younger fresh groundwaters and the deeper groundwaters, which have undergone a greater degree of water-rock interactions. This has produced a spatial pattern in $\delta^{30}$Si across the aquifer due to the local hydrogeology, resulting in a correlation between $\delta^{30}$Si and tritium activities when considering all groundwaters ($\rho = -0.68$, p = 0.0002). In the deeper aquifer, the inverse correlation between dSi and Cl concentrations ($\rho = -0.79$, p = 0.04) for the more saline groundwaters is attributed to groundwater mixing with local seawater that is depleted in dSi (<3.6 μM). Our results from this well-constrained, island aquifer system demonstrate that stable Si isotopes usefully reflect the degree of water-aquifer interactions, which is related to groundwater residence time and local hydrogeology. Our finding that lithogenic Si dissolution occurs in the freshwater lens and the freshwater-seawater transition zone on Rottnest Island appears to supports the recent inclusion of a marine submarine groundwater discharge term in the global dSi mass balance. Geologically-young, carbonate aquifers, such as Rottnest Island, may be an important source of dSi in coastal regions with low riverine input and low oceanic dSi concentrations.

## 1. Introduction

Dissolved silicon (Si(OH)$_4$; dSi) is a key nutrient in global biogeochemical cycles that is sourced primarily from continental silicate weathering (Tréguer et al., 1995;Tréguer and De La Rocha, 2013;Rahman et al., 2017;Rahman et al., 2019;Frings et al., 2016). Thus, the dSi flux from the continents is a key control on primary productivity in the global biogeochemical cycles of the oceans (Falkowski et al., 1998). Stable silicon isotopes ($^{28}$Si, $^{29}$Si and $^{30}$Si) are useful for tracing the rate and extent of silicate weathering reactions due to the preferential incorporation of lighter Si isotopes into during clay mineral formation (Frings et al., 2015;Pogge von Strandmann et al., 2012;Hughes et al., 2013;Georg et al., 2009a;Georg et al., 2007), silica precipitation (Geilert et al., 2014;Oelze et al., 2015), and adsorption of Si onto Fe-Al (oxy)hydroxides (Opfergelt et al., 2009;Opfergelt et al., 2017).

Biological processes also fractionate Si isotopes as dSi is utilised by organisms such as diatoms and vascular plants (Ding et al., 2008;De La Rocha et al., 1998;Meyerink et al., 2017). Moreover, Si isotopes are not fractionated during congruent mineral dissolution and there is a narrow range of $^{30}$Si values for the upper continental crust (UCC): -0.3 ± 0.2 ‰ (2 standard deviations; s.d.; Savage et al., 2013). Therefore, dSi isotopic ratios may reflect the balance between congruent silicate dissolution and secondary mineral formation such that if the global Si budget is well constrained, marine Si isotope records may be used to reconstruct past changes in continental weathering and primary productivity (Frings et al., 2016;De La Rocha et al., 1998;Christina et al., 2000).

The global dSi isotopic budget is poorly constrained for groundwater systems (Frings et al., 2016), despite around half of the total dissolved solids (TDS) flux to the oceans deriving from submarine groundwater discharge (SGD) (Zektser and Loaiciga, 1993). Globally, average dissolved $\delta^{30}$Si values in low-temperature groundwater systems (+0.2 ± 0.8 ‰) are lower than those in rivers (+1.3 ± 0.7 ‰; 1 s.d.; Frings et al., 2016). Similarly, groundwater fluxes from other key marine weathering proxies such as Mg, Ca, and Sr also exhibit distinct isotopic compositions from global riverine averages (Mayfield et al., 2021). The lower $\delta^{30}$Si values in some groundwater systems may be due to the greater extent of water-rock interactions compared to surface waters and the dissolution of $^{30}$Si-depleted secondary minerals (Basile-Doelsch et al., 2005;Georg et al., 2009b;Pogge von Strandmann et al., 2014). The $\delta^{30}$Si values reported for groundwater dSi are highly variable even within a single system, e.g., ranging from −0.15 to +1.34 ‰ at various depths in the alluvial Bengal Basin aquifer in India (Georg et al., 2009a), +0.4 to +1.0 ‰ for volcanic springs in Iceland (Opfergelt et al., 2011), −1.4 to +0.6 ‰ in a sandstone aquifer in Arizona, U.S. (Georg et al., 2009b), and −1.5 to −0.9 ‰ in the sedimentary Great Artesian Basin, Australia (Pogge von Strandmann et al., 2014). In a sandy coastal aquifer, a large gradient was found between the $\delta^{30}$Si values of the fresh groundwater (+1.0‰) and seawater (+3.0 ‰; (Ehlert et al., 2016). Although ~12% of the global SGD flux flows through carbonate aquifers (Beck et al., 2013), these systems have received little attention thus far. This is because carbonates are not expected to contain much Si; however, some carbonate-dominated aquifers contain copious amounts of silica-bearing material of various origins, e.g., alluvial, aeolian, pedogenic, etc. (Muhs, 2017).

Here we present Si isotopic compositions of groundwater from a coastal carbonate aquifer, Rottnest Island (RI), Australia. There is a freshwater lens on RI located above a ~10 m freshwater-seawater transition zone (Playford et al., 1977). Conventional stable isotope data ($^{2}$H, $^{13}$C, and $^{18}$O), tritium ($^{3}$H) and radiocarbon ($^{14}$C) measurements show that the freshwater lens aquifer on RI is recharged by modern rainfall, which fluctuates due to climatic variations, and that the residence time for fresh groundwaters ranges from ~12 to 36 years (Bryan et al., 2020). The stable isotope and major element (Mg, Ca) geochemistry of fresh RI groundwaters is dominated by carbonate weathering reactions (Bryan et al., 2017). Carbonate weathering processes are not expected to affect groundwater $\delta^{30}$Si values due to the low Si incorporation into the carbonate minerals, e.g. the dSi partition coefficient for precipitated calcite is ~0.001 (Hu et al., 2005), and the low Si/Ca ratio for the acid-soluble component of the aquifer bedrock (Martin et al., 2020). A biological influence on groundwater $\delta^{30}$Si values is probably not important for the RI fresh groundwaters because: 1) the area above the freshwater lens is sparsely vegetated due to land clearing and contains no surface water features, and 2) the salt lakes (located to the east of the freshwater lens) that host diatom communities are not hydrologically connected to the groundwater system (Bryan et al., 2016), and 3) there is no reported occurrence of biogenic opal in the Tamala Limestone (Semeniuk and Semeniuk, 2006;Hearty and O'Leary, 2008). In contrast, trace elements, such as strontium (Sr) and lithium (Li), are derived mainly from the dissolution of silicate minerals found within carbonate aeolianite matrix of the RI aquifer (Martin et al., 2020). Moreover, these fresh RI groundwaters are saturated with respect to quartz and their Li isotopic compositions suggest that water-aquifer interactions with secondary minerals occur in the shallow aquifer (Martin et al., 2020). Deeper in the aquifer on RI, there are saline groundwaters that have undergone a greater degree of seawater mixing Bryan et al. (2017). These saline RI groundwaters have lower $\delta^{7}$Li values relative to fresh RI groundwaters, suggesting that they

are interacting with the silicate basement rocks (Martin et al., 2020). Here we assess the application of Si isotopes as a weathering proxy in a carbonate-dominated aquifer with well-constrained hydrogeochemical parameters for fresh and saline groundwaters combined with bedrock data. This study presents the first Si isotope measurements in a carbonate island aquifer system and provides insights into subsurface weathering processes on a high spatial resolution scale. We also assess the potential contributions of carbonate-dominated aquifers to the global Si-isotope budget of the oceans.

## 2. Study area

The surface features, hydrogeology, climate and geology of Rottnest Island (RI) have been characterised by Bryan et al. (2016, 2017, 2020) and are summarised here. Briefly, RI is a ~19 km$^2$ island located 18 km from Perth, Australia. The maximum elevation on RI is ~45 m Australian Height Datum (AHD) (Fig. 1). European settlement on RI in the 1830s reduced the native vegetation cover, but revegetation has commenced on the island, except for the area above a freshwater lens located in the central part of the island to increase groundwater recharge there. Sand dunes are a common feature on RI and there is an absence of water courses. There are a number of hypersaline lakes at sea level (Playford and Leech, 1977), and lower-salinity waterbodies in interdunal wetlands on RI (Gouramanis et al., 2012).

The main lithology on RI is the Tamala Limestone, which is a ~115 m thick Late Quaternary carbonate aeolianite that lies unconformably upon Cretaceous fluvial sandstone beds (Playford et al., 1976). The Tamala Limestone deposit on RI forms part of the world's longest (~1,000 km) carbonate aeolianite formation along the western Australian coastline (Brooke, 2001). The silicate content of Tamala Limestone on RI ranges from 1-6 wt.% and mostly comprises quartz sand grains, which is representative of other aeolianites globally (Muhs, 2017). The carbonate minerals present in Tamala Limestone include aragonite, pure calcite, low-Mg calcite, and high-Mg calcite (Martin et al., 2020).

The climate on RI is Mediterranean with hot-dry summers, and mild-wet winters. The annual average rainfall (1880–2015 CE) is 691 mm and annual reference evapotranspiration is 1694 mm, but rainfall on RI has been below average rainfall since the 1960s. The majority of groundwater recharge occurs during large precipitation events in winter. The freshwater lens on RI is located above a ~10 m freshwater-seawater transition zone within the upper section of the Tamala Limestone (Playford et al., 1977). Since the late 1970s, the extent of the freshwater lens has decreased due to a decrease in rainfall, resulting in seawater intrusion into the freshwater lens (Bryan et al., 2016). Sea-level high stands (~2 m higher than present) at ~7 ka (Coshell and Rosen, 1994) and 4.4 ka (Gouramanis et al., 2012) would have probably intruded seawater into the groundwater system. Cations adsorbed onto the aquifer matrix during previous/ongoing seawater intrusion episodes may, therefore, be a source of dissolved solids to groundwaters through cation exchange processes.

The studied groundwaters were grouped into three mixing types on the basis of their depth and hydrogeochemical properties: fresh, transition zone 1 (T1), and transition zone 2 (T2) (Table A.1; Bryan et al., 2017). Fresh groundwaters are shallow (above -1 m AHD), have low TDS values (<1 g L$^{-1}$), Cl concentrations from ~3-8 mM and are expected to be younger (ca. <0.5 ka) with tritium ($^3$H) and radiocarbon ($^{14}$C$_{DOC}$) values of >0.6 tritium units (TU) and >89 percent modern carbon (pMC), respectively; moreover, lumped parameter modelling suggests that the mean residence time for fresh groundwaters ranges from ~12 to 36 years (Bryan et al., 2020). In contrast, the T2 groundwaters are the deepest studied here (-5 to -15 m AHD), have higher TDS (7 to 30 g L$^{-1}$), Cl concentrations from ~97-561 mM and estimated to be older (ca. 3-7 ka) with $^3$H and $^{14}$C$_{DOC}$ values of <0.3 TU and <67 pMC, respectively. The T1 groundwaters are located at depths between the fresh and T2 groundwaters (-1 to -5 m AHD) and have intermediate compositions due to mixing between fresh and T2 groundwaters caused by seasonal groundwater level fluctuations.

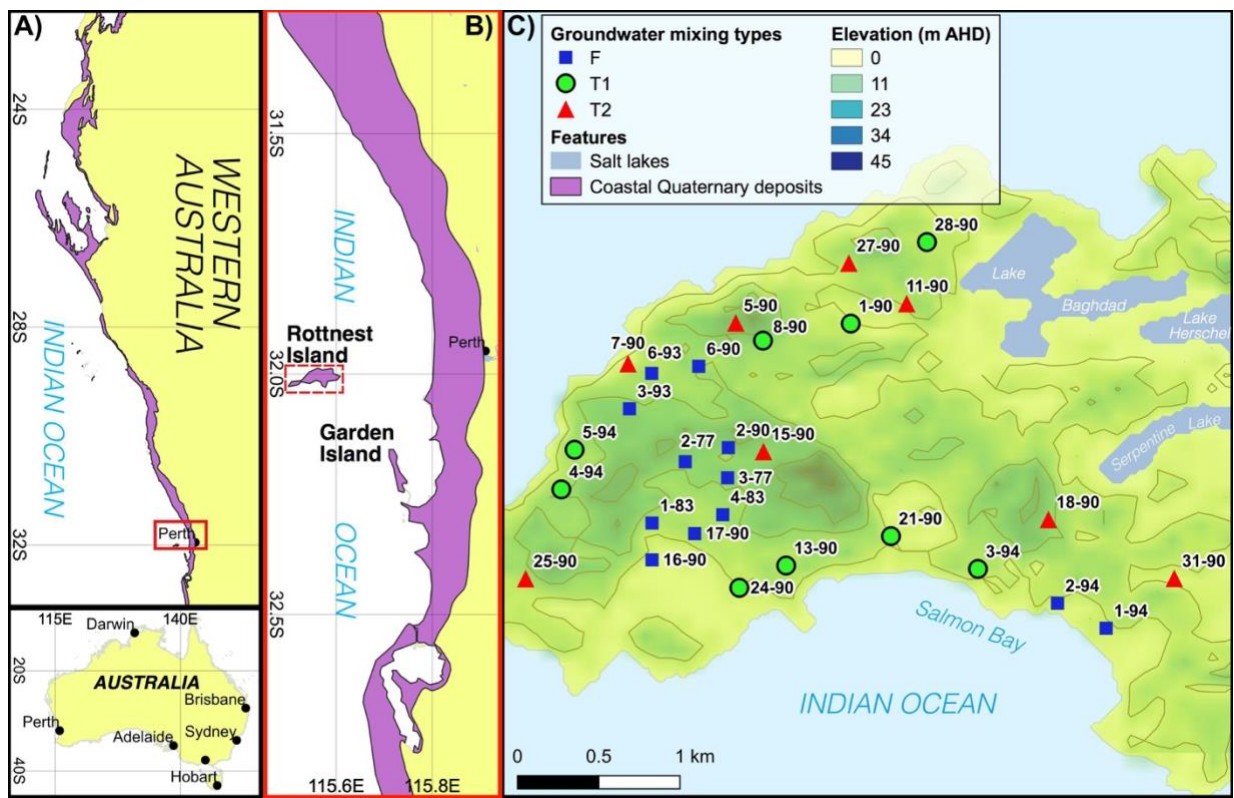

**Figure 1. Maps showing the locations of coastal Quaternary deposits in Western Australia (A and B) (Stewart et al., 2008); and C) a Digital Elevation Model of RI, Australia showing sampling locations and IDs (modified from Martin et al., 2020). Groundwater classification types are defined according to Bryan et al. (2017) and underlined sample IDs represent bores with available groundwater residence times modelled by Bryan et al. (2020). The red dashed line in C shows the location of the cross section shown in Fig. 2.**

## 3. Methods

### 3.1. Sample collection

The groundwater sampling protocols are described in detail by Martin et al. (2020). Briefly, 28 groundwater samples were collected from RI during two field campaigns in September 2014 and March 2015 (Fig. 1). Samples were collected at, or just above the well screen at the bottom of each well (maximum length = 1.5 m). Prior to sample collection, the monitoring and production wells were purged until in-field parameters stabilised. A seawater sample was collected from the shoreline using a peristaltic pump (Masterflex E/S portable sampler).

A rock and soil sample were collected from RI in March 2017. The rock sample (RI-B01) was sampled from an outcrop at Salmon Bay (32°00'46''S 115°30'33''E). Only visibly-unweathered material was sampled after removing weathered material using a geological hammer. The soil (RI-S01) was sampled from an unvegetated ridge with an actively-forming dune in the centre of RI, corresponding to the area above the freshwater lens (32°00'19''S 115°29'48''E; Fig. 1).

### 3.2. Analytical techniques

The concentrations of Si, other major elements, and selected trace elements used in this manuscript were determined by Bryan et al. (2016) and Martin et al. (2020) at the Australian Nuclear Science and Technology Organisation (ANSTO) by ion chromatography and inductively coupled plasma-atomic emission spectroscopy. The accuracy of cations and anions measurements were evaluated the charge balance error, with 80% of the samples falling within ±5% and all samples falling within ±6.2%, which is much greater than the uncertainty than the analytical precision of measurements.

Silicon isotope column chromatography procedures were conducted in clean laboratories at ANSTO. Acids used in all procedures were high-purity SEASTAR™ IQ grade hydrochloric acid (HCl) and nitric acid (HNO$_3$). These were further purified prior to use by sub-boiling using Savillex® PFA distillation apparatus and stored in Nalgene® FEP bottles. Dilutions were conducted using >18.2 MΩcm (at 25 °C) Milli-Q water. Acid concentrations were determined by titration with sodium hydroxide. Column chromatography was conducted using 1.8 mL Biorad AG 50W-X8 (100–200) cation exchange resin packed in Biorad ® 0.8 x 4 cm polypropylene microcolumns based on the one-step procedure from Georg et al. (2006). Prior to loading samples on columns, the resin and columns were cleaned with 3 mL of 3 M HCl, 6 M HCl, 7 M HNO3, 10 M HCl, 6 M HCl and 3 M HCl (18 mL acid in total) and conditioned with 6 mL of Milli-Q water. An appropriate volume of sample containing 7.2 μg of Si was loaded and eluted using Milli-Q water. Column calibration experiments ensured 100% Si yield, which was accomplished by elution with 4 mL of Milli-Q water. The Si fractions were collected in Savillex® PFA vials, evaporated to incipient dryness on a hotplate at 80°C and then re-dissolved in 2 mL of 2% (v/v) HNO$_3$, whilst monitoring for the formation of insoluble precipitates, which was not observed. This method yielded samples with a Si concentration of 3.6 ppm for isotopic analysis. As a secondary check on the column yield, the columns were eluted with a further 2 mL of Milli-Q water after collecting the Si fraction and acidified for screening during MC-ICP-MS analyses. Solid samples (RI-B01 and RI-S01), isotopic reference materials NBS-28 (quartz sand) and IRMM-018a (silica sand) were dissolved by alkali fusion to avoid using hydrofluoric acid. Approximately 10 mg of material was added to a platinum (Pt) crucible and the following reagents were added: 40 mg of LiBO$_2$ (Sigma Aldrich, ACS reagent, >98.0% purity) as a fluxing agent, and 100 mg of NaNO$_3$ (Sigma Aldrich, ReagentPlus®, >99.0%) as an oxidising agent. A procedural blank for the alkali fusion method was also prepared. This mixture was then placed in a pre-heated muffle furnace at 950°C using Pt-clad tongs for 40 min before switching off the furnace and left to cool overnight. The fusion cakes were then dissolved in 3 mL of 6 M HNO$_3$ and transferred to 50 mL centrifuge tubes and diluted to a Si concentration of ~100 ppm. The dissolved reference materials and blank solution were then processed through the column chemistry in the same manner as for samples.

The intensities of $^{28}$Si, $^{29}$Si and $^{30}$Si were analysed by multi-collector inductively-coupled plasma mass spectrometer (MC–ICP-MS) using a Thermo Scientific™ Neptune Plus™ in medium-resolution mode (M/ΔM ≈ 2000) at the Research School of Earth Sciences (RSES), Australian National University (ANU) following the analytical setup of Wille et al. (2010). Briefly, the MC-ICP-MS was operated in dry plasma mode using an ESI-Apex nebulizer with a Telfon inlet system, a demountable torch fitted with an alumina injector and using standard Ni sampler and skimmer cones. Prior to commencing sample measurements in each analytical session, the instrument was left to stabilise for 2-3 hours and the instrument tuning settings were optimised for signal stability and sensitivity (typically ~7 V/ppm). An external standard-sample-standard bracketing approach was employed to correct measured isotopic ratios for the mass-dependent sensitivity of Si isotopes during the MC–ICP-MS analyses (Albarède and Beard, 2004). Corrected Si isotopic ratios of samples are presented in ‰ as variations from quartz sand isotopic reference material NBS-28 ($\delta^{30}$Si$_{NBS-28}$ = +0.02) where $\delta^{29}$Si = [($^{29}$Si /$^{28}$Si)$_{sample}$ / $^{29}$Si/$^{28}$Si)$_{NBS-28}$− 1] × 1000 and $\delta^{30}$Si = [($^{30}$Si /$^{28}$Si)$_{sample}$ / $^{30}$Si/$^{28}$Si)$_{NBS-28}$− 1] × 1000. Silica sand IRMM-018a was analysed as an unknown standard and had a $\delta^{30}$Si value of -1.52 ± 0.12 ‰ (n = 5, 2 s.d.), which is within error to previously measured values (Ziegler et al., 2010;Baronas et al., 2018;Geilert et al., 2020). A plot of $\delta^{30}$Si vs $\delta^{29}$Si confirms that all samples plot on the mass-dependent fractionation line to yield a fractionation factor (β) of 0.516 ± 0.010 (Fig. A.1). Unfortunately, the uncertainty on β does not permit us to discriminate between kinetic and equilibrium fractionation, which have β values of 0.509 and 0.518, respectively (Frings et al., 2016). The total procedural blanks from alkali fusion and column chemistry, and acidified Milli-Q water samples eluted after collecting the Si fraction during column chemistry (to assess column yield) were assessed during the MC–ICP-MS analyses. The $^{30}$Si signal intensities for these solutions could not be

distinguished for the background measured in 2% (v/v) $HNO_3$, typically <130 mV compared to standard and sample intensities of ~25 V. Thus, the blank contribution was less than 0.3% and all Si loaded onto the columns was eluted in the Si fraction.

## 4. Results

The groundwater sampling (e.g., pH, TDS, DO), geochemical (e.g., Ca, Mg, Cl concentrations, $\delta^{18}O$, $\delta^{13}C$) and radiometric data ($^3H$, $^{14}C_{DIC}$ and $^{14}C_{DOC}$) for our samples were described in detail by Bryan et al. (2016) and Bryan et al. (2017), and are provided in Appendix A.2. Samples were classified as either fresh, T1 or T2-type groundwaters (Bryan et al., 2017) on the basis of their hydrogeochemical properties and these definitions are used throughout this manuscript. Isotopic compositions are phrased in terms of $\delta^{30}Si$, acknowledging that this is a mass-dependent effect on all Si isotopes.

The dSi concentrations of the RI groundwaters ranged from 64 to 196 µM (Table 1) with an average value of 117 µM. The average value on RI is slightly higher than the average for coastal carbonate aquifers globally (80 ± 63 µM, 1 s.d.; Rahman et al., 2019), but much lower than the average values for other coastal aquifers, e.g. igneous (604 ± 192 µM, 1 s.d.), and 'complex' mixed lithology aquifers (288 ± 245 µM, 1 s.d.; Rahman et al., 2019). Although the groundwater TDS values broadly increased with depth for the fresh (<1 g $L^{-1}$), T1 (1 to 5 g $L^{-1}$) and T2 groundwaters (>5 g $L^{-1}$) (Fig. 2a), there was no correlation between dSi concentrations and depth, or salinity, for the fresh and T1 groundwaters (Fig. 2b). However, when adopting a significance level threshold of p = 0.05, the deeper T2 groundwaters exhibited a significant inverse correlation between dSi concentrations and Cl concentrations ($\rho$ = -0.79, p = 0.04, n = 5; Fig. 3a). Moreover, the dSi concentrations of the T1 groundwaters (average: 133 ± 36 µM, 1 s.d.) were higher (p = 0.0002) than those of the fresh groundwaters (average: of 112 ± 23 µM, 1 s.d.) and the T2 groundwaters (average: 107 ± 43 µM, 1 s.d.; p = 0.001) according to Mann-Whitney U tests, which is a non-parametric test suitable for comparing for independent variables with small sample sizes. Although 13-90 was collected in a different field campaign (March 2015), there is no clear effect of seasonality on the Si data and this is not discussed further in the manuscript. Moreover, lumped-parameter modelling by Bryan et al. (2020) showed that the mean residence times of fresh groundwaters is ~40 years and no seasonal variations in groundwater geochemistry are to be expected.

The groundwater Al concentrations were <0.4 or <1.9 µM, except for 1-94 and 25-90 (1.5 and 30.4 µM, respectively, Table 1). Similarly, Fe concentrations were <0.9 µM for most groundwaters, except some T1- (21-90 and 24-90) and T2 groundwaters (11-90 and 15-90; Table 1). The groundwater Mn concentrations were <18 µM) for all fresh groundwaters and most T1 groundwaters (except 21-90 and 24-90), but higher Mn concentrations were found for the deeper T2 groundwaters (36 to 1,470 µM), except for 27-90 (<18 µM). The high Mn concentration for T2 groundwater 31-90 (1,470 µM) greatly differs from the average groundwater Mn concentration (98 ± 282 µM, 1 s.d.) and may be considered as an outlier. When excluding 31-90 as an outlier and fresh groundwaters with low Fe and Mn concentrations, there is a significant correlation Mn and Fe concentrations ($\rho$ = 0.82, p= .0004, n = 14).

The $\delta^{30}Si$ values of RI groundwaters ranged from -0.4±0.3 to +3.6±0.2 ‰ with an average of +1.6±0.9 ‰ (1 s.d., n = 28, Table 1). The $\delta^{30}Si$ values in the fresh RI groundwaters (+0.2±0.4 to +2.6±0.3 ‰) extended to a slightly higher range than those of some continental aquifers (+0.1 to +1.7 ‰; Georg et al., 2009a; Opfergelt et al., 2011), and there was an absence of very low $\delta^{30}Si$ values (less than -1.0 ‰), which have been found in some continental aquifers (Georg et al., 2009b; Pogge von Strandmann et al., 2014). In constrast to the wide range of $\delta^{30}Si$ values measured for the fresh RI groundwaters, fresh groundwaters from the quartz-rich, sandy, barrier island aquifer of Spiekeroog, northern Germany exhibited homogeneous $\delta^{30}Si$ values of +1.0 ± 0.2‰ (Ehlert et al., 2016). The $\delta^{30}Si$ values for the shallower fresh (average: +1.3±0.7 ‰, 1 s.d.) and T1 groundwaters (average: +1.3±0.9 ‰, 1 s.d.) could not be distinguished according to a Mann-Whitney U test (p = 0.749), however, the $\delta^{30}Si$ values for the deeper T2 groundwaters (average: +2.6±0.6 ‰, 1 s.d.) were resolvably higher than both the fresh groundwaters (p = 0.003), and the T1

groundwaters (p = 0.009). There was a statistically significant correlation between $\delta^{30}$Si values and Cl concentrations for the fresh RI groundwaters ($\rho = 0.73$, p = 0.008, n = 10), but not for the T1 and T2 groundwaters ($\rho = -0.64$, p = 0.11, n = 5; Fig. 3b). For the fresh groundwaters, $\delta^{30}$Si values correlated with dSi concentration ($\rho = 0.59$, p = 0.02, n = 9; Fig. 4a), but this correlation was not found for the T1 or T2 groundwaters (Fig. 4b). Interestingly, the fresh groundwater with the lowest $\delta^{30}$Si value (1-94; +0.2‰) also exhibited the highest Al concentration (1.5 µM) and, similarly, the T2 groundwater with the highest Al concentration (25-90; 30 µM) also had the lowest $\delta^{30}$Si value (+1.7‰). There were no correlations between the dSi and $\delta^{30}$Si values of RI groundwaters with Mn or Fe concentrations, respectively.

A previous study measured the $^{87}$Sr/$^{86}$Sr ratios and $\delta^{7}$Li values of the RI groundwaters (Martin et al. 2020). These values ranged from 0.709167 to 0.709258 (average: 0.709192, n = 19) and +14.5 ‰ to +36.3 ‰ (average: +29.8 ± 5.9 ‰, 1 s.d., n = 23; see Appendix A.2 for individual values), respectively. No correlations were found between the $\delta^{30}$Si values, $^{87}$Sr/$^{86}$Sr ratios and $\delta^{7}$Li values of the RI groundwaters

The Si concentrations of the rock (RI-B01) and soil (RI-S01) samples were 542 and 839 ppm, respectively, and their $\delta^{30}$Si values were -0.8±0.4 and -0.1±0.3 ‰, respectively (Table 2). The $\delta^{30}$Si value of RI-S01 is close to the average for the upper continental crust (UCC; -0.3 ± 0.3 ‰, 2 s.d.) (Savage et al., 2013), whereas RI-B01 was slightly below the average UCC value.

Table 1. dSi concentrations and isotopic ratios and groundwaters and a seawater sample from Rottnest Island

| ID | Groundwater type[a] | Sampling Date | Screen Elevation | Cl | DO | Al | Mn | Fe | dSi[b] | $\delta^{30}$Si | | 2 s.e. |
|---|---|---|---|---|---|---|---|---|---|---|---|---|
| (RI-) | | | (m AHD) | (mM) | (mg/L) | (µM) | (µM) | (µM) | (µM) | (‰) | | |
| 2-77 | F | 9/29/14 | -0.11 | 4.6 | 1.8 | <0.4 | <18 | <0.1 | 81.9 | +0.7 | ± | 0.4 |
| 3-77 | F | 9/29/14 | -0.01 | 5.1 | 1.5 | <0.4 | <18 | <0.1 | 89.0 | +0.7 | ± | 0.3 |
| 1-83 | F | 9/28/14 | 0.09 | 6.1 | 1.6 | <0.4 | <18 | <0.1 | 124.6 | +1.3 | ± | 0.5 |
| 4-83 | F | 9/30/14 | -0.11 | 4.4 | 4.2 | <0.4 | <18 | <0.1 | 113.9 | +1.0 | ± | 0.5 |
| 2-90 | F | 9/29/14 | -0.50 | 4.2 | 3.6 | <0.4 | <18 | <0.1 | 74.8 | +1.2 | ± | 0.5 |
| 6-90 | F | 9/29/14 | -0.64 | 6.8 | 3.4 | <0.4 | <18 | <0.1 | 135.3 | +2.3 | ± | 0.3 |
| 16-90 | F | 9/28/14 | -0.28 | 5.9 | 1.3 | <0.4 | <18 | <0.1 | 138.9 | +1.5 | ± | 0.6 |
| 17-90 | F | 9/30/14 | 0.06 | 5.9 | 3.5 | <0.4 | <18 | <0.1 | 121.1 | +1.1 | ± | 0.3 |
| 3-93 | F | 9/29/14 | -0.27 | 7.5 | 2.9 | <0.4 | <18 | <0.1 | 117.5 | +2.6 | ± | 0.3 |
| 6-93 | F | 9/29/14 | -0.22 | 5.5 | 4 | <0.4 | <18 | <0.1 | 131.7 | +1.6 | ± | 0.3 |
| 1-94 | F | 9/28/14 | -0.53 | 4.6 | 4.6 | 1.5 | <18 | <0.1 | 85.5 | +0.2 | ± | 0.4 |
| 2-94 | F | 9/28/14 | -1.00 | 2.7 | 7.1 | <0.4 | <18 | <0.1 | 128.2 | +1.0 | ± | 0.4 |
| 1-90 | T1 | 9/29/14 | -0.90 | 35.1 | 2.5 | <0.4 | <18 | <0.1 | 128.2 | +1.5 | ± | 0.5 |
| 8-90 | T1 | 9/29/14 | -0.59 | 9.2 | 1.9 | <0.4 | <18 | <0.1 | 192.3 | +2.8 | ± | 0.3 |
| 13-90 | T1 | 3/11/15 | -3.55 | 74.1 | 0.5 | <0.4 | <18 | <0.1 | NA | +1.2 | ± | 0.4 |
| 21-90 | T1 | 9/26/14 | -4.04 | 46.4 | 0.3 | <0.4 | 36 | 1.8 | 117.5 | +1.4 | ± | 0.3 |
| 24-90 | T1 | 9/26/14 | -3.47 | 11.9 | 0.4 | <0.4 | 91 | 2.0 | 131.7 | +1.2 | ± | 0.4 |
| 28-90 | T1 | 9/26/14 | -1.52 | 27.4 | 1.1 | <0.4 | <18 | <0.1 | 113.9 | -0.4 | ± | 0.3 |
| 3-94 | T1 | 9/27/14 | -0.72 | 13.7 | 3 | <0.4 | <18 | <0.1 | 78.3 | +1.8 | ± | 0.2 |
| 4-94 | T1 | 9/27/14 | -1.83 | 8.9 | 2.4 | <0.4 | <18 | <0.1 | 174.5 | +1.2 | ± | 0.4 |
| 5-94 | T1 | 9/27/14 | -1.87 | 8.9 | 1.1 | <0.4 | <18 | <0.1 | 128.2 | +0.6 | ± | 0.3 |

| 5-90 | T2 | 9/27/14 | -6.90 | 97.0 | 0.4 | <1.9 | 36 | <0.9 | 195.8 | +2.6 | ± | 0.2 |
|------|----|---------|-------|------|-----|------|-----|------|-------|------|---|-----|
| 11-90 | T2 | 9/26/14 | -6.19 | 473.2 | 0.5 | <1.9 | 146 | 3.3 | 81.9 | +3.6 | ± | 0.2 |
| 15-90 | T2 | 9/26/14 | -14.92 | 381.0 | 0.2 | <1.9 | 328 | 3.0 | 106.8 | +2.7 | ± | 0.4 |
| 18-90 | T2 | 9/27/14 | -11.16 | 409.1 | 0.3 | <1.9 | 146 | <0.9 | 64.1 | +2.8 | ± | 0.5 |
| 25-90 | T2 | 9/27/14 | NA | 225.0 | 0.3 | 30.4 | 55 | <0.9 | 99.7 | +1.7 | ± | 0.5 |
| 27-90 | T2 | 9/26/14 | -4.98 | 308.9 | 0.9 | <1.9 | <18 | <0.9 | 113.9 | +2.3 | ± | 0.4 |
| 31-90 | T2 | 9/28/14 | -9.20 | 244.2 | 0.3 | <1.9 | 1.470 | <0.9 | 85.5 | +2.1 | ± | 0.5 |
| SW-2 | SW | 29/9/15 | NA | 516.2 | 13.1 | <0.1 | <1 | <0.1 | <3.6 | NA | | |

[a]Defined by Bryan et al. (2017) (see Table A.1); [b]Measured by Martin et al. (2020).

**Table 2. Silicon concentrations, Si isotope ratios, and XRD data[a] for bulk rock samples from Rottnest Island**

| Sample | Description | Si | δ[30]Si | | 2 s.e. | Low-Mg Calcite ([Mg$_{0.03}$Ca$_{0.97}$]CO$_3$) | High-Mg calcite ([Mg$_{0.129}$Ca$_{0.871}$]CO$_3$) | Calcite (CaCO$_3$) | Aragonite (CaCO$_3$) | Quartz (SiO$_2$) | Sylvine, sodian |
|---|---|---|---|---|---|---|---|---|---|---|---|
| (RI-) | | ppm | ‰ | | ‰ | wt.% | wt.% | wt.% | wt.% | wt.% | wt.% |
| B01 | Rock | 542 | -0.8 | ± | 0.4 | 40.7 | 16.8 | 26.3 | 14.7 | 1.2 | 0.3 |
| S01 | Soil | 839 | -0.1 | ± | 0.3 | 28.1 | 52.7 | 0 | 13.2 | 6.0 | 0.0 |

[a]XRD data from Martin et al. (2020).

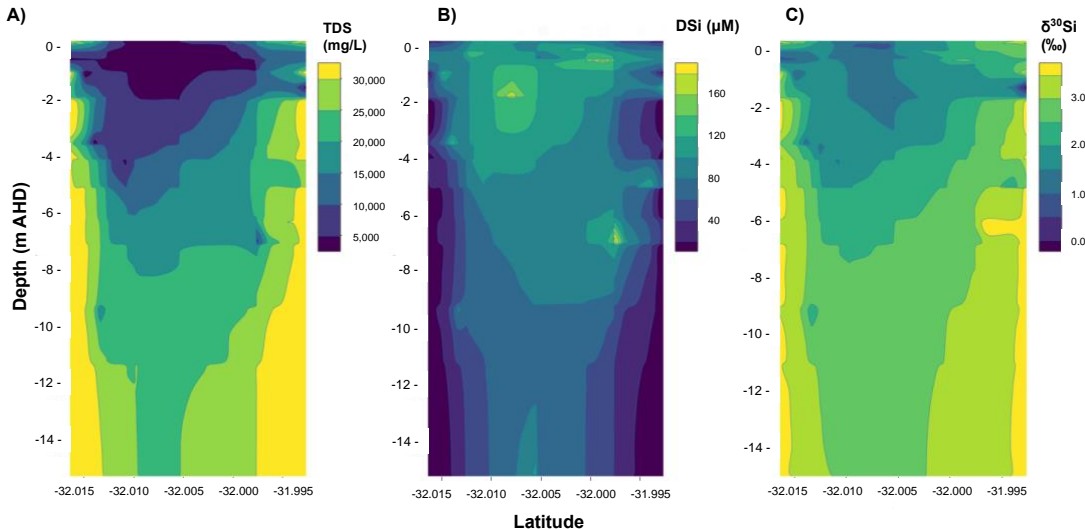

**Figure 2. (a) Gridded contour maps for a north-south transect of the aquifer Rottnest Island including data from all groundwater samples showing A) TDS values, B) dSi concentrations and C) δ[30]Si values for groundwaters as a function of depth and latitude in decimal degrees. The grids were created by interpolating groundwater data from their coordinates and depth using the loess function in the base package of the statistical programming language R, which is a non-parametric approach that fits multiple regressions within a given range.**

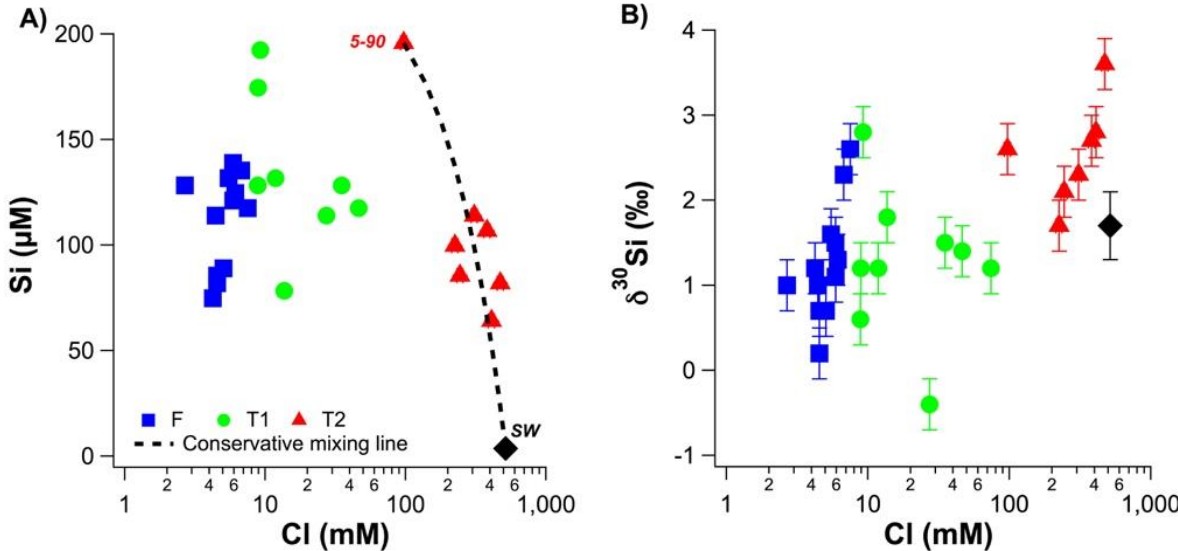

**Figure 3. A) Molar dSi concentrations as a function of molar Cl concentrations. B) δ³⁰Si values as a function of molar Cl concentrations. The dashed line in A) represents theoretical mixing between local seawater and the T2 groundwater with the highest dSi concentration (well 5-90).**

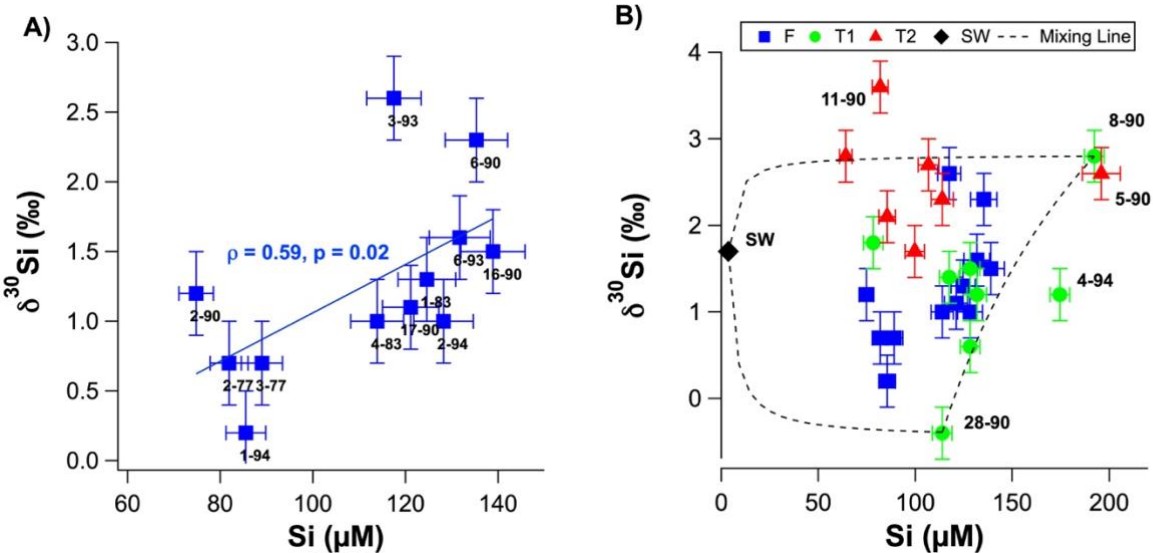

**Figure 4. A) Groundwater δ³⁰Si values as a function of molar dSi concentrations for fresh groundwaters; B) Groundwater δ³⁰Si values as a function of molar dSi concentrations T1 (green circles) and T2 groundwaters (red triangles). Black dashed lines represent theoretical mixing between seawater, well 8-90 and well 28-90. The seawater δ³⁰Si value was assumed to be +1.7‰ based on the average value for the low-concentration seawater sample from Grasse et al. (2017). Sample IDs are shown next to markers for all samples in A) and selected samples in B).**

## 5. Discussion

To understand the processes fractionating Si isotopes in the coastal aquifer on RI, we first attempt to identify the sources of dSi in the unsaturated zone for fresh meteoric groundwaters and then consider the effect of water-rock interactions on their dSi isotope composition. The effect of mixing processes in the deeper, older and more saline T1 and T2 groundwaters is then considered.

### 5.1. Identifying the source of dSi in meteoric groundwaters at Rottnest Island

The primary source of dSi in groundwater systems is often the dissolution of silicate minerals in the aquifer, i.e., lithogenic silica, but the positive relationship between dSi and δ³⁰Si ($\rho = 0.59$, p = 0.02) in fresh RI groundwaters suggests physical mixing and/or

diffusion processes must also be considered. An inverse relationship is typically found in hydrological systems in which dSi is derived from silicate mineral dissolution and removed by secondary mineral formation (Ehlert et al., 2016;Opfergelt et al., 2011;Georg et al., 2009a). The unusual positive relationship between dSi and $\delta^{30}Si$ on RI could be explained by a two-component mixing model, but this requires the low dSi/low $\delta^{30}Si$ and a high dSi/high $\delta^{30}Si$ end-members to be identified.

The low dSi/low $\delta^{30}Si$ end-member can be identified by the intercept of the regression line between dSi concentrations and $\delta^{30}Si$ (-0.7±0.9‰, 1 s.d.), which is similar to the $\delta^{30}Si$ values of the aquifer bedrock, Tamala Limestone (-0.8±0.4‰ to -0.1±0.3‰; Table 2). The primary host of dSi is likely the aquifer bedrock (Tamala Limestone), which has a quartz content ranging from 1.2 to 6.0 wt.% (Martin et al., 2020). The fresh groundwater with the highest Al concentration (1-94; 1.5 µM), also had the lowest $\delta^{30}Si$ value (+0.2‰), providing evidence for the dissolution of silicates other than quartz in the shallow aquifer. Another possibility is that dSi is supplied from the dissolution of secondary minerals, e.g., clay minerals and silcrete. Although it is difficult to distinguish detrital phases from secondary weathering products produced in-situ, lower $\delta^{30}Si$ values (less than -1 ‰) would be expected if the dissolution of secondary minerals was occurring on RI, as measured in the Navajo sandstone aquifer (Georg et al., 2009b) and the mixed sedimentary aquifer (alternating layers of sandstone, siltstone and mudstone) of the Great Artesian Basin, Australia (Pogge von Strandmann et al., 2014). On this basis, we conclude that the dissolution of silicate minerals is the primary source of dSi in fresh RI groundwaters.

The high dSi/high $\delta^{30}Si$ end-member could reasonably expected to be seawater or the numerous hypersaline salt lakes on RI, but a saline source is unlikely and a better candidate may be the T1 groundwaters. Firstly, both local seawater and the hypersaline lakes have very low dSi concentrations (<3.6 µM; Martin et al., 2020) compared to the fresh RI groundwaters (>64 µM; Table 1). Moreover, the Cl contents of the fresh groundwaters are low and correspond to estimated seawater fractions of less than 1% (Bryan et al., 2016). The similarly low dSi concentrations in rainfall (<3.6 µM; Martin et al., 2020) also render the evapoconcentration of rainfall as an important source of dSi in groundwaters as unlikely, especially considering the episodic nature of rainfall events on RI that recharge the aquifer and the resulting low effective rates of evaporation of rainfall (Bryan et al., 2020). Rather, we propose that the $\delta^{30}Si$ values of fresh groundwater may broadly reflect the degree to which they have undergone vertical mixing with older, more evolved T1 groundwaters from intermediate depths. This is supported by 1) the overlap in the distribution of dSi and $\delta^{30}Si$ values for the fresh and T1 groundwaters, i.e., most T1 groundwaters lie within one standard deviation of the average $\delta^{30}Si$ value for fresh groundwaters (Fig. 4b); 2) the correlation between the $\delta^{30}Si$ values and Cl concentrations of fresh groundwaters (Fig. 3b); and 3) the correlation between $\delta^{30}Si$ values and $^3H$ activities when considering all groundwater types (Fig. 5). Vertical mixing processes may explain the spatial pattern of groundwater $\delta^{30}Si$ and dSi values, for instance, the highest $\delta^{30}Si$ values are found in the northeastern portions of the freshwater lens (wells 6-90 and 3-93). Although these groundwaters are defined as "fresh" in terms of their TDS concentrations (<1 g $L^{-1}$), there is a greater degree of upward mixing of older, more saline groundwater from the freshwater-seawater transition zone in these wells (Bryan et al., 2020). In contrast, the lowest $\delta^{30}Si$ values are found in the central area of the freshwater lens (wells 2-77, 2-90 and 3-77) and SW areas of the freshwater lens (wells 1-94 and 2-94; Fig. 6). These groundwater wells typically receive more rainfall recharge and have the shortest groundwater residence times according to their $^3H$ activities (Bryan et al., 2020). Therefore, the high dSi/high $\delta^{30}Si$ end-member is identified as older groundwaters which are more evolved in terms of their dSi and $\delta^{30}Si$ composition. The specific role of water-rock interactions in meteoric groundwaters is discussed in Section 5.2.

## 5.2. The role of water-rock interactions in meteoric groundwaters at Rottnest Island

The increase of $\delta^{30}Si$ in fresh RI groundwaters from +0.2±0.4 to +2.6±0.3 ‰ may occur by the following mechanisms: 1) the incorporation of Si into neo-formed clay minerals (Frings et al., 2015;Georg et al., 2007;Hughes et al., 2013;Opfergelt et al.,

2017;Pogge von Strandmann et al., 2012), 2) amorphous silica precipitation (Oelze et al., 2015;Geilert et al., 2014;Opfergelt et al., 2017), and/or 3) the adsorption of Si onto Fe-Al (oxy)hydroxides (Opfergelt et al., 2009;Oelze et al., 2014). The enrichment of the heavier Si isotopes in fresh RI groundwaters by secondary mineral formation associated with incongruent weathering conditions is supported by PHREEQC modelling employing the "water4f.dat" database (Parkhurst and Appelo, 2013). These results indicate that most fresh RI groundwaters are saturated with respect to secondary minerals, such as kaolinite and montmorillonite (Fig. 7), and the increase of $\delta^{30}$Si may, therefore, be attributed to the precipitation of clay minerals. This is supported by the high $\delta^{7}$Li values of fresh RI groundwaters (>23 ‰), whereby $^{6}$Li is preferentially incorporated into secondary phases (Martin et al., 2020). Although Si and Li isotopes are both useful tracers of silicate weathering processes, there is no correlation between $\delta^{7}$Li and $\delta^{30}$Si in RI groundwaters, possibly highlighting that different processes control the isotopic fractionation of these elements in coastal aquifers. One key difference might be the high Li content of seawater relative to meteoric groundwaters, whereas dSi is a nutrient and depleted in local seawater. This is an important distinction on RI since modern seawater intrusion (Bryan et al., 2016), and past sea-level high stands (~2 m higher than present), e.g., events at ~4 and 7 ka (Coshell and Rosen, 1994; and Gouramanis et al., 2012), would have probably intruded seawater into the shallow groundwater system. Such seawater intrusion episodes would be expected to adsorb Li, but not Si, onto the aquifer matrix during previous/ongoing and provide Li to groundwaters through cation exchange processes. The importance of ion-exchange in fresh RI groundwaters is supported by a Mg excess and a deficit of Na and Ca relative to theoretical mixing with seawater (Bryan et al., 2017).

Fresh RI groundwaters are also saturated with respect to common Fe (oxy)hydroxide minerals, such as iron(III) hydroxide and goethite (Fig. 7). These minerals may bind dSi over a wide range of pH conditions (Fein et al., 2002), and preferentially adsorb the lighter Si isotopes (Oelze et al., 2014;Opfergelt et al., 2009), thereby increasing the $\delta^{30}$Si values of fresh RI groundwaters. In addition to freshly-precipitated Fe minerals, a pre-existing source of Fe-Al (oxy)hydroxide minerals in the aquifer may be the 'terra rossa' Late Quaternary paleosol units that occur throughout the Tamala Limestone (Smith et al., 2012;Hearty and O'Leary, 2008), and similar materials could occur distributed through the aquifer rather than being specifically the paleosols themselves. These paleosol units may be particularly important for determining preferential groundwater flow paths as they can act as inception horizons in the Tamala Limestone and it is likely that groundwaters will flow through or along paleosol units on RI (Hearty and O'Leary, 2008). As the enrichment of heavier Si isotopes in the aqueous phase depends on the degree of soil weathering, the Fe-oxide content, and the proportion of short-range ordered Fe-oxides (Opfergelt et al., 2009), the extent of Si isotopic fractionation in fresh RI groundwaters may depend on the specific geochemical properties of the paleosol units through which they flow and produce spatial variations in Si isotopic compositions.

Finally, amorphous silica precipitation appears to be unimportant on RI as many fresh groundwaters are undersaturated with respect to amorphous silica (Fig. 7). This is reasonable considering that amorphous silica is very soluble under ambient temperature and pressure conditions and only precipitates at dSi concentrations in excess of 2 mM (Gunnarsson and Arnórsson, 2000). As this threshold concentration is an order of magnitude higher than those in fresh RI groundwaters (Table 1), it is unlikely that amorphous silica precipitation occurs in fresh RI groundwaters.

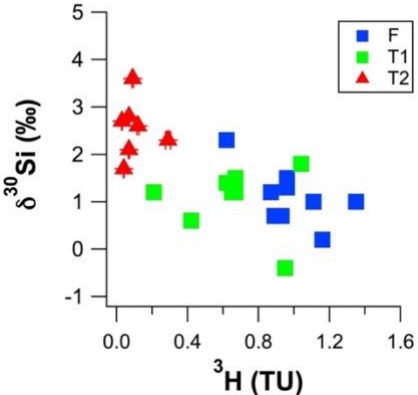

**Figure 5 δ³⁰Si values for RI groundwaters as a function of their ³H concentrations (in tritium units; TU)**

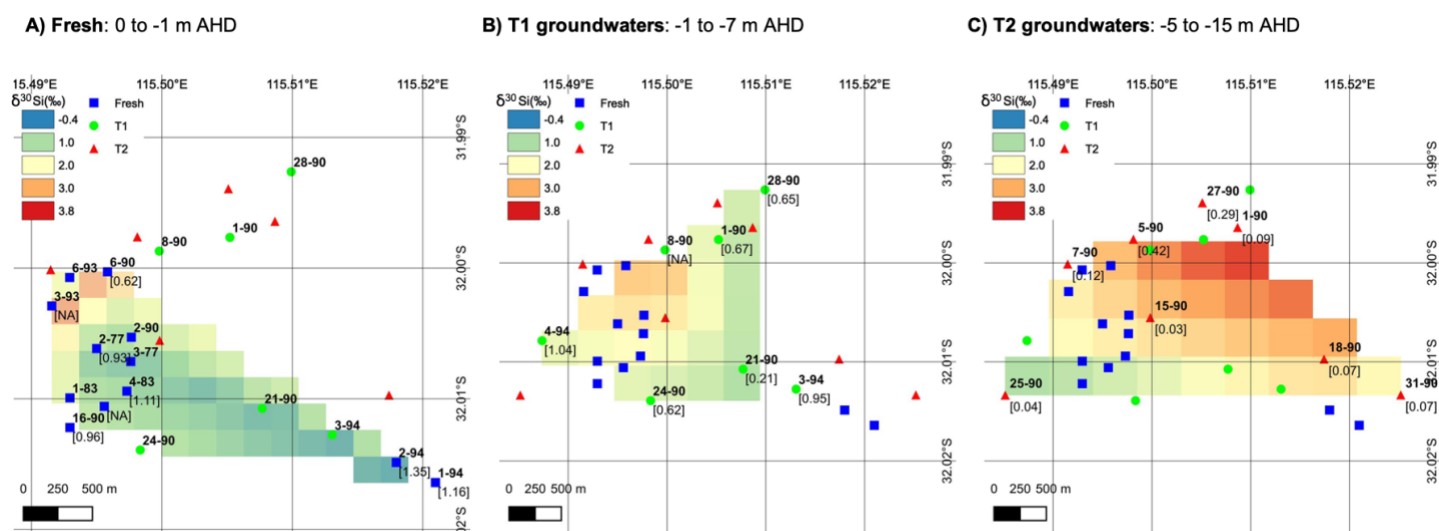

Figure 6 Interpolated spatial plots of δ³⁰Si values for groundwaters at RI for the A) fresh, B) T1 and C) T2 groundwaters. The data were created using a TIN interpolation function (cubic interpolation method) with a pixel size of ca. 0.002 (A) to 0.003° (B and C), depending on sample density per unit area. Sample IDs are shown in bold text next to markers and values in square brackets show ³H values from Bryan et al. (2016).

330

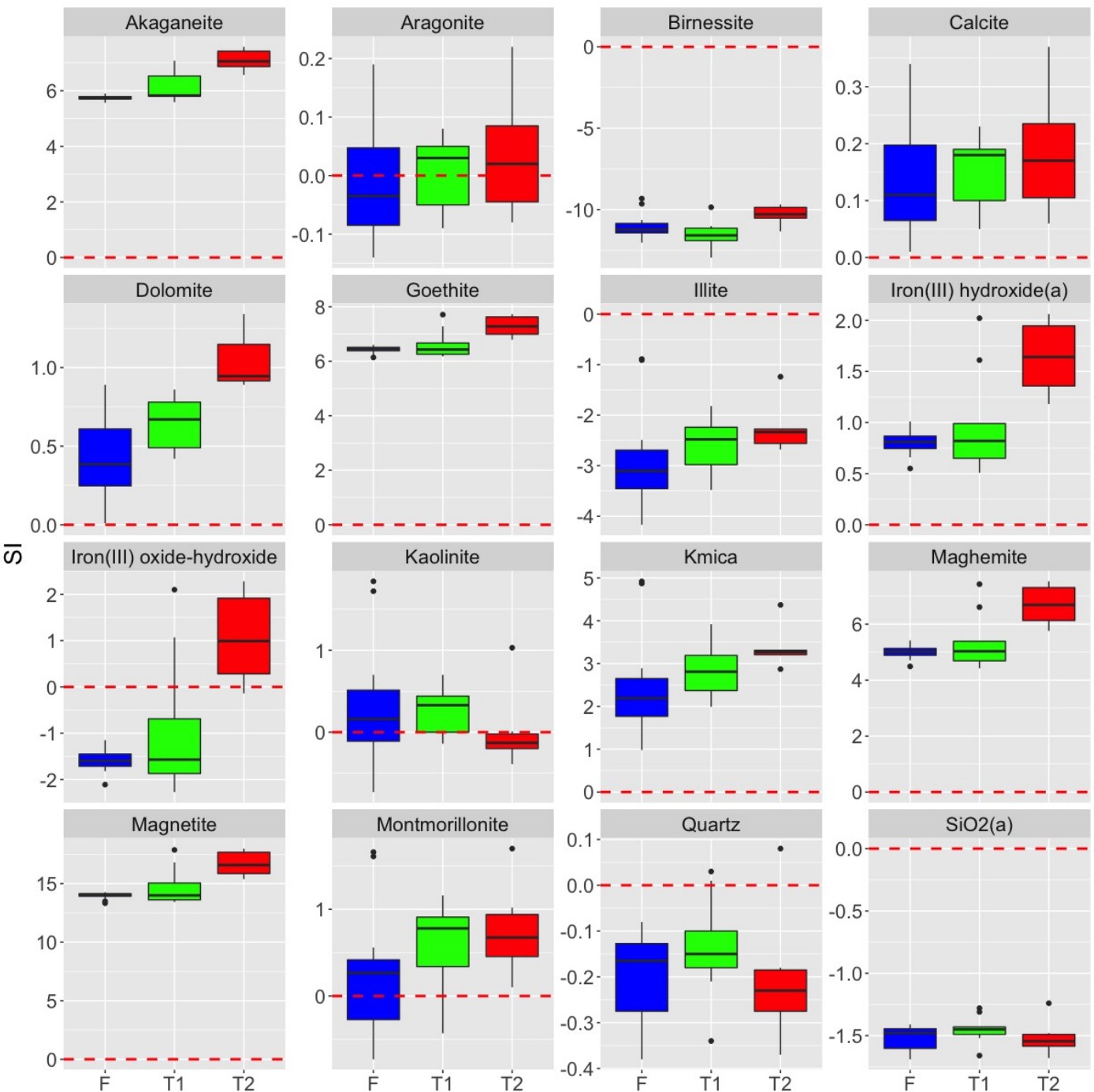

**Figure 7 Box plots showing the saturation indices of selected minerals grouped by groundwater mixing type (F: fresh, T1, and T2). Dashed red lines show SI = 0, and black dots represent outliers (1.5 times greater than the interquartile range).**

### 5.3. Mixing processes in the freshwater-seawater transition zone at Rottnest Island

Groundwaters in the freshwater-seawater transition zone on RI can be divided into two water types on the basis of their chemical and isotopic composition as T1 or T2 groundwaters (Bryan et al., 2017). The T2 groundwaters are the deeper, older, and more saline, whereas the T1 groundwaters have compositions that are dominated by mixing between the fresh and the more saline T2 groundwaters. This mixing is driven by tidal and groundwater level fluctuations within the transition zone and variable meteoric recharge (Bryan et al., 2016).

The $\delta^{30}$Si values of T1 groundwaters may be used to establish a three-component end-member mixing model between seawater (low dSi, high $\delta^{30}$Si), groundwaters that have undergone less extensive water-aquifer interactions (low dSi, low $\delta^{30}$Si, e.g., well 28-90), and groundwaters that have undergone more extensive water-aquifer interactions (high dSi, high $\delta^{30}$Si, e.g., well 8-90). The low $\delta^{30}$Si value well 28-90 (-0.4±0.3 ‰, Table 1) could be explained by the dissolution of Tamala Limestone (-0.8 to -0.1 ‰, Table 2) in this groundwater whilst not undergoing secondary mineral formation, or interacting with paleosol units, or any other phases that may preferentially adsorb Si. In contrast, the high $\delta^{30}$Si value for well 8-90 (+2.8 ‰) is associated with the highest dSi concentration of all T1 groundwaters (192 μM) suggesting that, at this location in the freshwater-seawater transition zone, the dissolution of silicate minerals is accompanied by enhanced secondary mineral formation, or Si adsorption onto secondary phases, e.g. with paleosol units.

The more saline T2 groundwaters in the deeper aquifer on RI have high seawater fractions (>17%; Bryan et al., 2017), whereby their Si isotopic composition is primarily controlled by mixing with local seawater that is depleted in dSi (<3.6 μM; Table 1). This mixing process dilutes the dSi concentrations in the T2 groundwater and explains the negative correlation between dSi and Cl ($\rho$ = -0.79, p = 0.04, n = 7). The higher $\delta^{30}$Si values for the deeper T2 groundwaters compared to the fresh and Tl groundwaters may be attributed to the high $\delta^{30}$Si value of local seawater. Although this was not measured in our study, the groundwater with the highest seawater fraction (84%, well 11-90) also has the highest $\delta^{30}$Si value (+3.6±0.2 ‰, Table 1). Although the $\delta^{30}$Si value of local seawater was not measured, it's dSi concentration was very low and a value of around +1.7‰ is expected according to the well-established, inverse relationship between dSi and $\delta^{30}$Si in seawater (Singh et al., 2015;Grasse et al., 2013;Grasse et al., 2017). Although all T2 groundwaters plot broadly on a mixing line between seawater (<3.6 μM dSi) and the well with highest groundwater dSi concentration (well 5-90, Fig. 3a), there is also evidence for lithogenic silica dissolution in the deeper aquifer. For instance, the T2 groundwater with the lowest $\delta^{30}$Si (25-90; +1.7 ‰), also has a much lower $^3$H value (0.04 TU). Such low groundwater $^3$H are consistent $^{14}$C data indicating that there is older (>3 ka) seawater 'trapped' under RI in the deeper aquifer (Bryan et al., 2017). Additional evidence for ongoing weathering reactions in the deeper aquifer is the $\delta^7$Li value of the T2 groundwater from the deepest well exhibiting the lowest $\delta^7$Li value measured on RI (Martin et al., 2020). Lower $\delta^7$Li (and $\delta^{30}$Si) values relative to fresh RI groundwaters are consistent with either continued lithogenic silica dissolution of the aquifer bedrock (Tamala Limestone) or interactions with the silicate basement rocks (Martin et al., 2020). Thus, the extent of $\delta^{30}$Si variations is primary controlled by mixing with seawater but there is clear evidence for continued dissolution of silicate minerals in the deeper aquifer on RI.

**5.4.    Implications for the global dSi isotope mass balance**

Our finding that the dissolution of lithogenic silica is occurring in the freshwater-seawater transition zone supports the inclusion of a 'marine SGD' term in the global dSi mass balance that represents the dissolution of lithogenic silica coastal sediments by recirculated seawater (Rahman et al., 2019;Cho et al., 2018). This is consistent with the findings from the quartz-rich, sandy, barrier island aquifer of Spiekeroog, northern Germany (Ehlert et al., 2016), and supported by the significantly higher dSi concentrations of the T1 groundwaters than both fresh (p = 0.0002) and T2 groundwaters (p = 0.001, Table 1). Marine SGD accounts for the marine-derived component of dSi in SGD that is supplied from the lithogenic dissolution of coastal sediments by recirculated seawater. Previous mass balance models only considered terrestrial SGD inputs ('fresh SGD') with a dSi flux of ~0.7 Tmol a$^{-1}$ (Frings et al., 2016;Tréguer and De La Rocha, 2013), which equates to ~10% of the riverine dSi flux. Marine SGD is estimated to supply ~3.1 Tmol a$^{-1}$ of dSi globally (Rahman et al., 2019). The total SGD flux (fresh SGD + marine SGD) is ~3.8 Tmol a$^{-1}$ representing ~60% of the riverine dSi input (~6.3 Tmol a$^{-1}$) and is six-fold higher than previously recognised. As highlighted by Rahman et al. (2019), the major effect of the revised dSi global mass balance is the large decrease in the estimated dSi oceanic residence time from ~10 to ~8 ka.

The Si isotopic composition of the fresh SGD flux is poorly constrained, and our data provide one of the few direct estimates of this component. This is because RI is a coastal island aquifer that receives no terrestrial runoff and the contribution of dSi from lithogenic silica dissolution in the more saline RI groundwaters can be estimated using our data from the fresh RI groundwaters. As the average $\delta^{30}$Si values for the fresh and T1 groundwaters were similar (+1.3±0.7 ‰ and +1.3±0.9 ‰, respectively; 1 s.d.; Table 1), a value of +1.3 ‰ may be adopted for the fresh SGD flux. Adopting this value for the fresh SGD flux in a revised dSi mass balance decreases the estimated ocean-average $\delta^{30}$Si value from +0.8 to +0.1 ‰. A value of around +2.6 ‰ for the marine SGD term is required to be consistent with the current estimate of the ocean-average $\delta^{30}$Si value, which corresponds to the $\delta^{30}$Si value of the saline groundwater with the highest dSi concentrations on RI (well 5-90). These estimates are highly uncertain and highlight the need for additional Si isotopic measurements of saline groundwaters to obtain a representative estimate of the marine SGD term and better constrain the global Si isotopic budget.

Our measurements may also be used to estimate the continental dSi flux in Western Australia as Rottnest Island forms part of the longest carbonate aeolianite unit in the world, the Tamala Limestone. The terrestrial SGD flux from Western Australia is estimated to be 3.2 km$^3$/a with carbonate aquifers comprising 50% of the coastline (Zekster et al. 2007; Rahman et al., 2019). By adopting the average dSi concentration of the fresh groundwaters (112±23 μM), we estimate that the dSi flux from the Tamala Limestone is 1.79 x10$^8$ ± 0.37 x10$^8$ mol/a of dSi, corresponding to ~27% of the dSi flux from Western Australia. Whilst this value is insignificant relative to the estimated global dSi flux of 7.1 x10$^{11}$ mol/a, it is significant on a continental scale. Moreover, carbonate aeolianite aquifers are an important feature of many coastlines globally, e.g., The Yucatan, South Australia, The Bahamas, and should be considered as important sources of dSi in these regions.

## 6.    Conclusions

The Si isotopic composition of groundwaters was utilised to trace groundwater processes on RI. We find that the main source of dSi in fresh RI groundwaters is the dissolution of primary silicate minerals present within the aquifer bedrock, Tamala Limestone. The $\delta^{30}$Si values of fresh groundwaters on RI appears to be related to the incorporation of the lighter Si isotopes into secondary minerals formed in the aquifer, but amorphous silica precipitation does not appear to be important. Higher $\delta^{30}$Si values were found for wells with a greater degree of upward mixing with older, more saline groundwater from the freshwater-seawater transition zone, whereas lower $\delta^{30}$Si values were found in groundwater wells that typically receive more rainfall recharge and have the shortest groundwater residence times. Thus, the local hydrogeology of RI may explain the spatial pattern in groundwater $\delta^{30}$Si values across the freshwater lens Therefore, the stable Si isotopic composition of groundwaters provides useful information on the degree of water-aquifer interactions, which is expected to increase with the time elapsed since meteoric recharge.

The main source of dSi in shallow, fresh groundwaters on RI was the dissolution of lithogenic silica found within the carbonate matrix of the aquifer rock. Moreover, this process appears to continue in more saline groundwaters in the freshwater-seawater transition zone, which has been termed the 'marine SGD flux'. Constraining the Si isotopic composition of this flux from key areas, such as volcanic island aquifers, should be a goal of future studies to better constrain the global Si isotopic budget. Although not important on a global scale, the dissolution of lithogenic silica in coastal carbonate aeolianite aquifers may be an important source of oceanic dSi regionally. For instance, the local seawater around RI is depleted in dSi and two orders of magnitude lower than fresh RI groundwaters. Thus, we propose that the coastal carbonate aeolianite aquifers may be an important source of oceanic dSi in other coastal regions with carbonate aquifers where there is no large dSi riverine flux and the oceanic concentration of dSi is low, e.g., The Bahamas or the Yucatán Peninsula, Mexico.

**Code and data availability**

The data in this manuscript are available from the authors upon request.

**Author contribution**

AB, KM and MN conceptualised the research. KM and EB collected the water samples, and AM and AB collected the rock and soil samples. AM conducted the analytical work with assistance from MN. AM prepared the manuscript with contributions from all co-authors.

**Competing interests**

The authors declare that there are no competing conflicts of interest.

**Acknowledgements**

The Rottnest Island Authority (RIA) is thanked for supporting this project, especially C. Thomas, L. Wheat and S. Kearney. We also recognise the enduring backing of S. Hollins, Head of Research at ANSTO, in addition to H. Wong and C. Vardanega for supporting ICP-MS and ICP-OES analyses, C. Dimovski and S. Hankin for field trip preparations, and K. Simon and D. Child for usage of the clean laboratory facilities. Les Kinsley at ANU is acknowledged for sharing his wisdom and support in MC-ICPMS analyses. This study was supported by an ARC Linkage grant (LP150100144).

**A.1 Appendix**

Supporting tables and figures are provided in this Appendix. All geochemical parameters presented in this manuscript are given in Table S1.

**A.1.1 Potential effects of anions during Si isotopic measurements**

Our Si column chromatography procedure utilised a cation exchange resin following Georg et al. (2006), which is effective for separating Si from cations, but has the disadvantage that anions such as $SO_4^{2-}$ may be eluted with the Si fraction. This could cause matrix effects between $SO_4$-free bracketing standards and $SO_4$-containing samples during MC-ICP-MS measurements due to mass bias and induce an offset to measured Si isotopic ratios (Hughes et al., 2011;van den Boorn et al., 2009), which is also shown by the positive correlation between the measured intensities of the Si isotopes with $SO_4/Si$ and $\delta^{30}Si$ (van den Boorn et al., 2009). This effect does not appear to be important for fresh meteoric samples with lower $SO_4^{2-}$ contents (Georg et al., 2006), but may be important for highly saline samples. As a brucite co-precipitation method was not adopted for our samples, it should be assessed whether the $SO_4$ present in groundwaters affected our Si isotopic measurements, however, it should also be noted that we only processed meteoric groundwaters that had sufficiently high dSi and low TDS as to not exceed 10% of the resin capacity and, thus, the $SO_4/Si$ molar ratios of analysed groundwaters were at least ~30-fold lower than for local seawater, which had a molar $SO_4/Si$ ratio of  ~9,000.

In our groundwater analyses, there was no correlation between the $SO_4/Si$ molar ratios and the measured signal intensity for $^{29}Si$ (before dilution to match bracketing standards) during MC-ICPMS measurements (Fig A.2A), or $\delta^{30}Si$ values and $SO_4/Si$ molar

ratios (Fig A.2B). Although it might be argued that there is a weak correlation for the T2 groundwaters between the $SO_4/Si$ molar ratios and signal intensity for $^{29}Si$ ($R^2 = 0.33$; Fig A.2A) and $\delta^{30}Si$ values ($R^2 = 0.27$; Fig A.2B), these weak correlations are not statistically significant (p-values <0.05). Moreover, there was no correlation between the measured signal intensity for $^{29}Si$ and $\delta^{30}Si$ values for all samples (Fig A.2C). We, therefore, conclude that matrix effects caused by anions in Si solutions did not induce a detectable offset for $\delta^{30}Si$ values. As noted by by Hughes et al. (2011), the importance of the anionic matrix effect is likely laboratory-specific due to the various combinations of different instruments, sample matrices, analytical settings, and/or Si column chromatography.

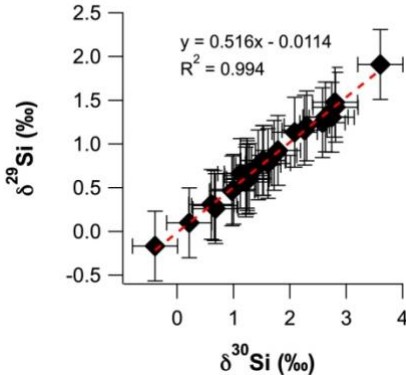

**Figure A.1 Linear plot of $\delta^{29}Si$ as a function of $\delta^{30}Si$ for all groundwater samples**

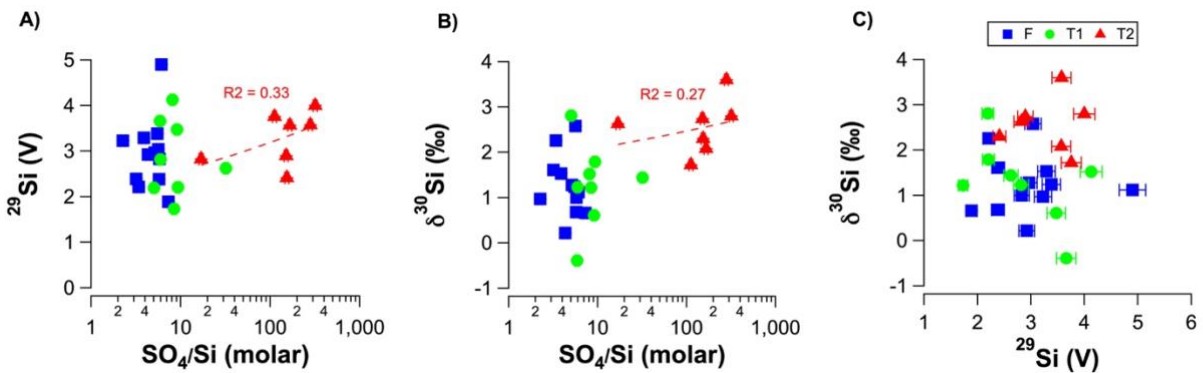

**Figure A.2 Linear plots for RI groundwater samples showing A) signal intensity for $^{29}Si$ measured during MC-ICPMs measurements (before dilution to match bracketing standards) as a function of molar $SO_4/Si$ ratios, B) $\delta^{30}Si$ as a function of molar $SO_4/Si$ ratios, and C) $\delta^{30}Si$ as a function of signal intensity for $^{29}Si$ measured during MC-ICPMs measurements (before dilution to match bracketing standards). The dashed red lines in panels A and B show the regression lines for the T2 groundwaters.**

**Table A.1 Groundwater mixing types (Bryan et al., 2017)**

| Mixing type | Depth (m AHD) | Cl (mM) | TDS (g L$^{-1}$) | $^{3}$H (TU) | $^{14}$C DOC |
|---|---|---|---|---|---|
| Fresh | 0 to -1 | 2.7-7.5 | <1 | 0.6-1.4 | 89-105 |
| T1 | -1 to -7 | 8.9-320.9 | 1-21 | 0.2-1.0 | 84-98 |
| T2 | -5 to -15 | 97.0-560.6 | 7-30 | 0.0-0.3 | 47-67 |

**Table A.2 Groundwater residence times from Bryan et al. (2020) and corresponding Si isotope data**

| ID | Min | Max | Average | dSi | δ³⁰Si |
|---|---|---|---|---|---|
| | (a) | (a) | (a) | (μM) | (‰) |
| 1-83 | 15.0 | 61.6 | 40.9 | 124.6 | 1.3 |
| 16-90 | 12.0 | 64.0 | 39.4 | 138.9 | 1.5 |
| 4-83 | 11.5 | 52.4 | 37.4 | 113.9 | 1.0 |
| 6-90 | 26.9 | 63.6 | 53.3 | 135.3 | 2.3 |
| 3-77 | 17.8 | 64.9 | 44.8 | 89 | 0.7 |

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
