# Peer review of "The evolution of stable silicon isotopes in a coastal carbonate aquifer, Rottnest Island, Western Australia"

_Hydrology and Earth System Sciences, 2020_

## Referee Comment (RC1) · Anonymous Referee #1 · 15 Nov 2020

This manuscript attempts to explore Si isotope dynamics of groundwater collected from Rottnest Island, Australia to quantify dissolved Si flux and its isotope composition ($\delta^{30}$Si) in the marine SGD component from coastal limestone aquifers. Authors claim that Si released by dissolution of lithogenic materials (paleosols) followed by its adsorption on Fe-Al (oxyhydroxides) is the key mechanism fractionating Si isotopes in shallow and relatively fresh groundwaters, while deep samples are affected by seawater mixing. Thus, Si isotope compositions of the sampled groundwater reflect the degree of aquifer-water interaction depending on groundwater residence time and local hydrogeology. The dataset has major analytical and interpretation issues.

Accuracy and precision of Si isotope measurements are ensured from $\delta^{30}$Si of -1.52 ± 0.12 per mil determined for IRMM-018a (n=5). However, for accurate Si isotope results, each sample is analyzed generally a minimum of three times as blank-standard-blank-sample-blank-standard and the average $\delta^{30}$Si composition along with uncertainty (2s) is reported, which is not the case here. Given a single measurement for each sample, higher uncertainty of the order of 0.3-0.5 per mil is expected.

Samples are collected in two different seasons (Sep. 2014 and Mar 2015) but there is no mention of seasonality on the measured data.

Neither the reduction potential nor DO/Fe-Mn redox pairs are shown to reject the dissolution of Fe-Mn-Al oxyhydroxides in groundwater.

Labels are wrong in Fig. 1c (1-90 repeated) and Fig. 5. Labels as mentioned somewhere in the text do not match with Table 1.

Dissolved Si and $\delta^{30}$Si of fresh groundwaters show a significant positive correlation, which indicates a dominating physical mixing/diffusion control on the Si isotope budget also supported by the narrow spatial extent of groundwaters collected from vertical depths <1 m only dispersed over 1-2 km. Overlapping wide variations in dissolved Si and $\delta^{30}$Si of groundwater from intermediate depths further corroborates this idea.

Leaching experiments of rock and surface soil samples with 0.5 M HCl do not account for the preferential Ca dissolution from carbonates as silicate dissolution is a much slower process.

Fig 2a is redundant. Club spatial distributions of TDS and dissolved Si in Fig. 5

Fig. 3b: Low $\delta^{30}$Si in deep and saline groundwaters than expected from the seawater mixing (not shown) between 5-90 and seawater suggests in situ release of lithogenic Si (low $\delta^{30}$Si).

Fig. 5 High and low $\delta^{30}$Si values are seen for groundwater collected near and away from the Salt lakes located on eastern side of the island. This hints at the probable mixing between a significant in situ release of lithogenic Si (low $\delta^{30}$Si) and low Si (high $\delta^{30}$Si) in seepage waters of the salt lakes. The salt lake waters and sediment porewaters should also be measured.

Fig. 6: It has been used to show older groundwaters contain high $\delta^{30}$Si due to more preferential removal of lighter isotopes on Fe-Al oxyhydroxides. However, the groundwater 25-90 has the lowest $\delta^{30}$Si (1.72) and much lower $^3$H (0.04) among older deep water. This misleading figure contradicts Fig. 3b indicating in situ release of lithogenic Si (low $\delta^{30}$Si).

Section 5.4: The additional source of Si (high $\delta^{30}$Si) from the salt lakes excluded in this study can potentially bias dissolved Si isotope composition for the marine SGD component.

Table A2. Replace this table with Box plots of saturation indicies calculated for all samples and include Fe-Mn-Al minerals.

Suggestions and editorial corrections:

Do not club multiple parenthesis.

Conservative mixing is misleading due to in situ Si release and thus change it to theoretical mixing.

Line 118: delete space after "stabilized"

Line 131: add cm after "18.2 megaOhm"

Line 147: add Si after 28 and 29

Line 285: Delete "and northwestern"

Line 327 and elsewhere: change "rock-water" to "aquifer-water"

---

## Referee Comment (RC2) · Anonymous Referee #2 · 17 Dec 2020

Review of manuscript HESS-2020-429 submitted to Hydrology and Earth System Sciences by Martin and colleagues: The evolution of stable silicon isotopes in a coastal carbonate aquifer, Rottnest Island, Western Australia

Martin et al present silicon isotope ratios (expressed as $\delta 30Si$) from fresh and saline groundwater samples from a carbonate island in Western Australia. They interpret variation in $\delta 30Si$ of the freshwater samples to be driven by sorption of Si to Fe/Al oxides, while the saline water samples are governed by mixing with a low [Si], high d30si endmember. Given a recent focus on topics like e.g. boundary exchange, coastal filtering, SGD, etc. for the trace element and isotope budgets of the ocean, this paper

has potential to be an interesting case study.

In general, the manuscript is well written and seems appropriately referenced. The topic is an interesting one, I think, though does seem to fall slightly outside the scope of HESS as I understand it. I have some methodological concerns (detailed below), and some comments about the interpretations, some more major than others. I list these below in order they appear in the manuscript.

Comments

L71: this suggestion of interaction with basement rocks is interesting but never returned to. If it can be relevant for Li, why not for Si?

L95: relevance?

L100, 103: spell out units here – TU = tritium units? pMC = percent modern carbon? G PES = ??

L104 vs. L98 – repetition, but with age expectations reversed. Not sure what's going on here.

L127: Some indication of precision and accuracy needed here.

L132: This column protocol works sufficiently well for samples with low anionic components in the matrix. But for the saline samples, anions like sulphate or chlorine would elute at the same time as the sample. This could cause matrix effects between bracketing NBS28 standards and samples. Previous work has shown that matrix effects, sensu lato, can induce large bias to silicon isotope ratios (e.g. Hughes et al. 2011 JAAS. What steps were taken to correct for this, or demonstrate that it is not a problem? Typically, a pre-concentration step like the so-called 'MAGIC' protocol is used for brackish and marine samples, since this also has the effect of removing much of the matrix.

L137: Drying to 'incipient' dryness is also slightly worrying. What does this mean, in

practice? If silica precipitates because it becomes oversaturated in the solution, there is a danger that it will not redissolve in 2% HNO3. And because the precipitation is likely associated with a fractionation, this may also induce bias. Did the authors demonstrate 100% Si yield at this stage of protocol?

L140 or around: Give details of procedural blanks, for column chemistry and for alkali fusion

L147: Give 'Si' for 28 and 29.

L148: M/$\Delta$M = 2000 is probably at the low end of resolution with which the polyatomic interferences can be avoided, and could induce some noise to the measurement if some of the interference peaks overlap onto the measurement plateau (though the three-isotope plot shows this isn't a large problem – uncertainties should be given in this plot).

L155: This implies only 1 bracket was measured – is this correct? Normally, 3-5 individual standard-sample brackets are analysed per sample. If this is correct, the stated $\pm 0.12$ uncertainty may be too precise.

L158: There are more recent values for IRMM-018 – see e.g. Geilert et al. 2020 (Nat Comms); Baronas et al. 2018 (EPSL), etc.

L161: Not sure the uncertainties of $\pm 0.000$ are required here.

Figure 2: Panel A and B – the colorbar scale seems to imply the concentrations are negative. Figure 2: It's not clear which data are included in this. There is a red line in Fig 1, but presumably the samples falling off this line have been incorporated somehow. More detail could be useful.

L172: I find the conclusion that T1 DSi is higher than fresh and T2 very hard to believe given Fig. 3A. I am also skeptical about the correlation on L171.

L191: extra ")"

Results section: I can't see the Si concentration and $\delta 30Si$ for the seawater sample mentioned on L114. In e.g. Fig 4B, a literature value is used from Singh et al., but this is not sufficiently justified.

L262: I disagree that there is a threshold of Al concentration necessary for Si isotopes to fractionate – pure $SiO_2$ experiments also show fractionation (as noted elsewhere in this manuscript).

L264: If the interpretation for d7Li is the same as for $\delta 30Si$, why is there no relationship between the two? (L189)

L265: the evidence for Fe oxides playing an important role seems to be circumstantial. Is there any direct evidence they are present, with sufficient sorption capacity, to be important?

Section 5.2: There is no real explanation for the positive trend between Si concentration and isotope ratios – this is a bit counter-intuitive. One might expect a negative correlation between Si concentration and isotope ratio.

L305: missing word (divided?)

L312: See also above – this does not appear to be the case from Fig 3A.

Section 5.3.1. I struggle to follow the rationale for including 'T1' groundwaters as a separate case rather than just viewing them as intermediate between freshwaters and the marine/T2 samples.

L335 and around – the challenge seems to be to define the non-marine endmember. If this was more rigorously achieved, then the discussion might be more usefully structured in terms of deviations from conservative mixing as in e.g. estuary papers (Zhang et al. 2020 GCA). In general, it's also not clear where the marine endmember comes from. I don't think a value from the Bay of Bengal is necessarily very representative for waters bathing western Australia (see e.g. Holzer et al. 2015 GBC)

L355: Could/should this not instead yield low $\delta$30Si values if there is a re-equilibration between solute and rock?

L372: I don't understand this – wouldn't these values be representative of the terrestrial component of SGD?

It is disappointing there is no attempt to calculate fluxes and place them in a global context. In general, the global implications section is rather weak and seems added on as an afterthought. It would be good to see a better discussion of how the lessons from this case study can (or cannot) be applied elsewhere, for example, and a more rigorous attempt at upscaling.

―――――――――――――――

---

## Referee Comment (RC3) · Anonymous Referee #3 · 8 Feb 2021

The manuscript is well written, providing a critical dataset in an understudied system. I enjoyed reading it. The figures are generally clear. I have suggested some minor revisions and have some suggestions for the discussion. There were some few grammatical errors. I don't anticipate the revisions would take the authors very long to incorporate.

Suggested changes throughout the paper: I prefer "bSi" to "BSi" as the capital "B" in "BSi" can be mistaken for the symbol for Boron. Similarly, I suggest changing "DSi" to "dSi" so that the capital D won't be mistaken for Deuterium. For readers versed in the biogenic silica literature, the capital letters may not pose a problem. However, for new

readers it may contribute to some confusion.

Well written introduction. Clear reasoning, concise review, and compelling set-up and motivation for the project to fill a known data gap. I suggest adding Ehlert et al, 2016 (Ehlert, C., Reckhardt, A., Greskowiak, J., Liguori, B.T., Böning, P., Paffrath, R., Brumsack, H.J. and Pahnke, K., 2016. Transformation of silicon in a sandy beach ecosystem: Insights from stable silicon isotopes from fresh and saline groundwaters. Chemical Geology, 440, pp.207-218) to the 2nd paragraph (lines 42-54). That data was not included in Frings et al 2016.

2. Study Area The last paragraph is confusing w.r.t. description of "fresh" mixing type. One set of parameters is given for this zone in lines 99-101, and another set is given later in the paragraph (lines 103-105), with some repeated conditions (e.g., above 1m AHD). Also would it be possible to add a salinity or [Cl-] range to the three mixing zones? 3. Methods: No salt corrections or formation of a brucite precipitate was conducted to account for matric effects prior to purification on via Biorad AG 50W-X8? 4.Results Do the authors have a surface coastal water/open ocean value for dSi? I think the manuscript would be improved by adding a comparison of d30Si found in fresh-T2 to Ehlert et al 2016 in the second paragraph (lines 176-186). Incidentally, have the authors seen Mayfield et al., 2021 (Nature Communications)? Line 190: missing punctuation at end of sentence Line 191: extraneous parenthetical after "respectively" Fig. 2: So cool! Regarding the legend for TDS and dSi: are those negative values? Fig. 4: Can authors add a regression line to 4a as this is discussed in results and discussion? Also, in 4b, there are 2 green dashed lines and only description of 1 green dashed line in the caption.

5.1 Evidence for the source of DSi... Lines 223-226: referring to a figure here would be helpful Would you expect lower d30Si if RI was behaving like a closed system which could be described via Rayleigh distillation model? To clarify, if almost all the dSi supplied by quartz dissolution formed a secondary clay mineral, would the subsequent dissolution of that secondary mineral would impart a dissolved d30Si signature more

like the primary mineral? This scenario could be added to the discussion of secondary mineral dissolution in lines 230-235), unless the authors have other evidence this is not likely. Line 237: regarding bSi solubility, coastal settings can actually make for recalcitrant diatom frustules: proximity to dissolved Al and metals can help preserve frustules (e.g., solubility studies by Van Cappellen, Van Bennekom, and Loucaides et al 2012). I suggest authors clarify this sentence with specific conditions of their study site or leave out the last phrase about preservation in coastal sediments. Could the authors add some site specific data w.r.t. to dissolved Fe concentrations in their discussion of adsorption being the likely mechanism that takes up light isotope of dSi. What were the concentrations of Fe in the 0.5M HCl leachate? Are there dissolved Fe or O2 profiles at the sites? Are these sediments oxic? The more positive d30Si of dSi observed in groundwater in Ehlert et al 2016 was due to secondary mineral formation. Can the authors provide more evidence of why that's not occurring here? PHREEQC doesn't encompass the solubilities of all the amorphous-type alumino-silicates that can form. Line 286: refer to a figure Line 305: missing a word or phrase here "...can be into two water types..." Line 325-328: This three end-member mixing concept is important: could it be highlighted in one of the figures and accompanied by a reference to that figure at the end of this sentence? Or is this in Figure 4b?

Conclusions highlight broad impacts, well written.

---

## Author Comment (AC1) · 14 Mar 2021

**Author Response to Reviewers**

14 March 2021

Dear Editor,

We thank the three anonymous reviewers for taking the time to provide feedback in these challenging times and for greatly improving the manuscript due to their valuable input. We would like to take the opportunity to respond to these comments from all three reviewers in this document.

Please find the reviewer comments (RC) below in ***bold italics*** and our author responses (AR) in plain text. Where necessary, we have quoted the appropriate line numbers (LN) in the original or revised manuscript (MS). Note that at the request of Reviewer 3, we have changed "BSi" to "bSi" as the capital "B" in "BSi" can be mistaken for the symbol for boron. Similarly, "DSi" has been changed to "dSi" so that the capital "D" won't be mistaken for deuterium. This notation is also adopted in this response.

Best regards,

*Ashley Martin*

Dr. Ashley N. Martin (on behalf of all co-authors)

**RC1. This manuscript attempts to explore Si isotope dynamics of groundwater collected from Rottnest Island, Australia to quantify dissolved Si flux and its isotope composition (δ30Si) in the marine SGD component from coastal limestone aquifers. Authors claim that Si released by dissolution of lithogenic materials (paleosols) followed by its adsorption on Fe-Al (oxyhydroxides) is the key mechanism fractionating Si isotopes in shallow and relatively fresh groundwaters, while deep samples are affected by seawater mixing. Thus, Si isotope compositions of the sampled groundwater reflect the degree of aquifer-water interaction depending on groundwater residence time and local hydrogeology. The dataset has major analytical and interpretation issues.**

The reviewer has kindly summarised the findings in our MS, but unfortunately feels the dataset has major analytical and interpretation issues. We have addressed these concerns by responding to the specific comments outlined in this review and, where necessary, making the appropriate changes to the revised MS. We feel that these revisions have substantially improved the MS and would like to thank the reviewer for providing their insights.

**RC2. Accuracy and precision of Si isotope measurements are ensured from δ30Si of -1.52 ± 0.12 per mil determined for IRMM-018a (n=5). However, for accurate Si isotope results, each sample is analyzed generally a minimum of three times as blank-standard-blank-sample-blank-standard and the average δ30Si composition along with uncertainty (2s) is reported, which is not the case here. Given a single measurement for each sample, higher uncertainty of the order of 0.3-0.5 per mil is expected.**

As the reviewer stated, the accuracy and precision of our analyses were evaluated by repeat measurements of the reference material IRMM-018a, which were within error of reference values. We fully agree that it would have been preferable to measure samples in triplicate. However, resources allocated for analyses in the project were unfortunately limited by budget and time constraints. Following the reviewer's suggestion, we now include the internal uncertainty (2 standard error; 2SE) of the $\delta^{30}Si$ measurement for each sample in Table 1 of the revised MS and this is provided below. We apologise that these uncertainties were not included in the original submission; this was an oversight of the first author. The average uncertainty (2SE) of all sample measurements was 0.39‰, which is in line with the reviewers' estimated uncertainty of 0.3-0.5‰ for our analytical protocol. Although a lower level of uncertainty would be preferable, an uncertainty of ca. 0.4‰ does not affect our interpretations due to the wide range of silicon isotopic ratios measured in groundwaters at Rottnest Island (-0.39 to +3.60‰), which is

an order of magnitude higher than the uncertainty (ca. 4 vs 0.4‰). To better reflect this level of uncertainty, we now report Si isotopic data to only 1 decimal place in the revised MS. The error bars in all relevant figures in the revised MS have been updated to reflect the 2SE uncertainties.

**Table 1 (revised)**

| ID (RL-) | Water type[a] | Sampling date | Screen (m AHD) | Cl (mM) | DO (mg/L) | Al (uM) | Mn (uM) | Fe (uM) | dSi[b] (uM) | $\delta^{30}Si$ (‰) | | 2 s.e. |
|---|---|---|---|---|---|---|---|---|---|---|---|---|
| 2-77 | F | 9/29/14 | -0.11 | 4.6 | 1.8 | <0.4 | <18 | <0.1 | 81.9 | +0.7 | ± | 0.4 |
| 3-77 | F | 9/29/14 | -0.01 | 5.1 | 1.5 | <0.4 | <18 | <0.1 | 89.0 | +0.7 | ± | 0.3 |
| 1-83 | F | 9/28/14 | 0.09 | 6.1 | 1.6 | <0.4 | <18 | <0.1 | 124.6 | +1.3 | ± | 0.5 |
| 4-83 | F | 9/30/14 | -0.11 | 4.4 | 4.2 | <0.4 | <18 | <0.1 | 113.9 | +1.0 | ± | 0.5 |
| 2-90 | F | 9/29/14 | -0.50 | 4.2 | 3.6 | <0.4 | <18 | <0.1 | 74.8 | +1.2 | ± | 0.5 |
| 6-90 | F | 9/29/14 | -0.64 | 6.8 | 3.4 | <0.4 | <18 | <0.1 | 135.3 | +2.3 | ± | 0.3 |
| 16-90 | F | 9/28/14 | -0.28 | 5.9 | 1.3 | <0.4 | <18 | <0.1 | 138.9 | +1.5 | ± | 0.6 |
| 17-90 | F | 9/30/14 | 0.06 | 5.9 | 3.5 | <0.4 | <18 | <0.1 | 121.1 | +1.1 | ± | 0.3 |
| 3-93 | F | 9/29/14 | -0.27 | 7.5 | 2.9 | <0.4 | <18 | <0.1 | 117.5 | +2.6 | ± | 0.3 |
| 6-93 | F | 9/29/14 | -0.22 | 5.5 | 4 | <0.4 | <18 | <0.1 | 131.7 | +1.6 | ± | 0.3 |
| 1-94 | F | 9/28/14 | -0.53 | 4.6 | 4.6 | 1.5 | <18 | <0.1 | 85.5 | +0.2 | ± | 0.4 |
| 2-94 | F | 9/28/14 | -1.00 | 2.7 | 7.1 | <0.4 | <18 | <0.1 | 128.2 | +1.0 | ± | 0.4 |
| 1-90 | T1 | 9/29/14 | -0.90 | 35.1 | 2.5 | <0.4 | <18 | <0.1 | 128.2 | +1.5 | ± | 0.5 |
| 8-90 | T1 | 9/29/14 | -0.59 | 9.2 | 1.9 | <0.4 | <18 | <0.1 | 192.3 | +2.8 | ± | 0.3 |
| 13-90 | T1 | 3/11/15 | -3.55 | 74.1 | 0.5 | <0.4 | <18 | <0.1 | NA | +1.2 | ± | 0.4 |
| 21-90 | T1 | 9/26/14 | -4.04 | 46.4 | 0.3 | <0.4 | 36 | 1.8 | 117.5 | +1.4 | ± | 0.3 |
| 24-90 | T1 | 9/26/14 | -3.47 | 11.9 | 0.4 | <0.4 | 91 | 2.0 | 131.7 | +1.2 | ± | 0.4 |
| 28-90 | T1 | 9/26/14 | -1.52 | 27.4 | 1.1 | <0.4 | <18 | <0.1 | 113.9 | -0.4 | ± | 0.3 |
| 3-94 | T1 | 9/27/14 | -0.72 | 13.7 | 3 | <0.4 | <18 | <0.1 | 78.3 | +1.8 | ± | 0.2 |
| 4-94 | T1 | 9/27/14 | -1.83 | 8.9 | 2.4 | <0.4 | <18 | <0.1 | 174.5 | +1.2 | ± | 0.4 |
| 5-94 | T1 | 9/27/14 | -1.87 | 8.9 | 1.1 | <0.4 | <18 | <0.1 | 128.2 | +0.6 | ± | 0.3 |
| 5-90 | T2 | 9/27/14 | -6.90 | 97.0 | 0.4 | <1.9 | 36 | <0.9 | 195.8 | +2.6 | ± | 0.2 |
| 11-90 | T2 | 9/26/14 | -6.19 | 473.2 | 0.5 | <1.9 | 146 | 3.3 | 81.9 | +3.6 | ± | 0.2 |
| 15-90 | T2 | 9/26/14 | -14.92 | 381.0 | 0.2 | <1.9 | 328 | 3.0 | 106.8 | +2.7 | ± | 0.4 |
| 18-90 | T2 | 9/27/14 | -11.16 | 409.1 | 0.3 | <1.9 | 146 | <0.9 | 64.1 | +2.8 | ± | 0.5 |
| 25-90 | T2 | 9/27/14 | NA | 225.0 | 0.3 | 30.4 | 55 | <0.9 | 99.7 | +1.7 | ± | 0.5 |
| 27-90 | T2 | 9/26/14 | -4.98 | 308.9 | 0.9 | <1.9 | <18 | <0.9 | 113.9 | +2.3 | ± | 0.4 |
| 31-90 | T2 | 9/28/14 | -9.20 | 244.2 | 0.3 | <1.9 | 1.470 | <0.9 | 85.5 | +2.1 | ± | 0.5 |
| SW-2 | SW | 29/9/15 | NA | 516.2 | 13.1 | <0.1 | <1 | <0.1 | <3.6 | NA | | |

**RC3.** *Samples are collected in two different seasons (Sep. 2014 and Mar 2015) but there is no mention of seasonality on the measured data.*

We agree that the Si isotopic ratios of samples may vary seasonally. However, the majority of the samples (27 out of 29) were collected in September 2014 and only two samples were collected in

March 2015 to supplement the dataset (7-90 and 13-90). As only two samples were collected in March 2015 and do not contribute significantly to the overall dataset, we did not discuss the role of seasonality in the original MS. To address the reviewers concern, we removed data for 7-90 from Table 1 as no dSi or $\delta^{30}Si$ were measured for this sample. The Si isotopic data for 13-90 has been kept in the revised MS as this sample does not appear to represent an outlier in relation to the other groundwaters and does not influence the discussion. The sample collection dates were already presented in Table 1 of the MS. In addition, we now state that one sample was collected in a different field campaign in the revised MS.

Another point to consider is that the mean residence times of fresh groundwaters, for which lumped-parameter modelling was conducted by Bryan et al. (2020), is around 40 years (see Table A.3 in the original MS). Therefore, any seasonality is likely to be dampened by the multi-annual groundwater residence times. As the groundwater residence time increases with depth, any effects of seasonality will be even less relevant for the deeper T1 and T2 groundwaters.

***RC4. Neither the reduction potential nor DO/Fe-Mn redox pairs are shown to reject the dissolution of Fe-Mn-Al oxyhydroxides in groundwater.***

We agree with the reviewer that including these data would be useful for assessing the importance of Fe-Mn-Al oxyhydroxide dissolution in groundwaters. The DO data were actually already included in the supplementary information file, but we understand that this may not have been clear enough to the reader. The DO data were not included in the main MS as they were previously described by Bryan et al. (2016) and the authors wanted to ensure the originality of data presented in this MS and not repeat previous publications. The location of these data in the supplementary information file is now clearly stated in the revised MS. The DO concentrations of fresh, T1 and T2 groundwaters ranged from 1.3-7.1, 0.3-3.3, and 0.2-0.9, respectively. This suggests that the aquifer may become more anoxic with depth. As a detailed description of the Rottnest Island aquifer was already provided by Bryan et al. (2016), we had attempted to avoid purposefully characterising fundamental aquifer processes in this MS. As we now see the need for additional such descriptions, particularly with relation to redox processes, this is now discussed in the revised MS.

The groundwater Al, Mn and Fe concentrations were not included in the original MS as they were generally below the limit of detection (LOD) and difficult to interpret on this basis.

However, we now acknowledge that these data are of interest and should nevertheless be included in the manuscript. The Fe, Mn and Al concentrations are now included in Table 1 (given in Table 1 above in response to RC2) and are presented in the results section in the revised MS. In all fresh groundwaters except groundwater 1-94, the Al, Mn and Fe concentrations were below the LOD. Interestingly, 1-94 also had the highest Al concentration (1.5 µM) and the lowest $\delta^{30}$Si value (+0.2‰) of all the fresh groundwaters, which is now stated in the revised MS. The higher Al concentration and lower $\delta^{30}$Si value may suggest an increased role for silicate dissolution in the sample and this is now discussed in the revised MS.

In T1 groundwaters, the Al, Mn and Fe concentrations were below the LOD in all samples except 21-90 and 24-90, which had Mn concentrations of 36 and 91 µM, respectively, and Fe concentrations of 1.8 and 2.0 µM, respectively. However, the dSi and $\delta^{30}$Si values of these groundwaters are similar to other T1 groundwaters and do not appear to exhibit distinctive signatures for Fe-Mn oxyhydroxide dissolution. This is now discussed in the revised MS.

The T2 groundwaters had much higher Mn concentrations up to 1,470 µM, but generally low Al and Fe concentrations. The high Mn concentration for T2 groundwater 31-90 (1,470 µM) greatly differs from the average groundwater Mn concentration (98 ± 282 µM, 1 s.d.) and may be considered as an outlier. However, as groundwater 31-90 also had the lowest $^{13}C_{DIC}$ and $^{13}C_{DOC}$ values, the elevated Mn may be related to the redox cycling of Mn by microbial processes. There is no correlation between Mn concentrations and dSi or $\delta^{30}$Si values for the T2 groundwaters This is now discussed in the revised MS. Groundwater 25-90 had the highest Al concentration (30 µM) and the lowest $\delta^{30}$Si value (+1.7‰). This is now discussed in the revised MS. This may suggest an increased role for silicate dissolution in this deeper part of the deeper aquifer.

***RC5. Labels are wrong in Fig. 1c (1-90 repeated) and Fig. 5. Labels as mentioned somewhere in the text do not match with Table 1.***

We thank for the reviewer for bringing to our attention the error in Fig 1c. This error appears to result from overlapping label issues in the GIS software. An updated version of Fig 1c is now provided in the revised MS.

The Sample IDs in Table 1 were listed in the format RI_xx-xx, but did not have the 'RI_' precursor in the text. In the revised MS the Sample IDs in Table 1 are not listed without the 'RI_' precursor for consistency with the main text (as given in Table 1 above in response to RC2).

**RC6.  *Dissolved Si and δ30Si of fresh groundwaters show a significant positive correlation, which indicates a dominating physical mixing/diffusion control on the Si isotope budget also supported by the narrow spatial extent of groundwaters collected from vertical depths <1 m only dispersed over 1-2 km.***

The reviewer raises an interesting point about the potential role of diffusion in modulating the groundwater $\delta^{30}$Si values of fresh on Rottnest Island. However, this hypothesis requires the identification of a high dSi/high $\delta^{30}$Si end-member. This could be either local seawater or water from the RI salt lakes, but the dSi concentrations of these waterbodies are very low (<3.6 µM), which is an order of magnitude lower than the fresh groundwaters. This suggests that there is dSi is sourced from within the aquifer, which we identify to be the dissolution of lithogenic silica. We have attempted to further clarify this point in the revised MS.

One might also contend that we had already discussed the role of physical mixing and diffusion in Section 5.2 in LNs 284-297. Specifically, we highlighted that fresh groundwaters may be undergoing vertical mixing with older, more $\delta^{30}$Si-evolved T1 groundwaters. We have attempted to further clarify this point in the revised MS.

**RC7. *Overlapping wide variations in dissolved Si and δ30Si of groundwater from intermediate depths further corroborates this idea.***

As outlined above in our previous response (AR6), it might be argued that we already raised this point in our discussion in relation to vertical mixing of fresh groundwaters with older, more $\delta^{30}$Si-evolved T1 groundwaters. To further argue this, we now further discuss this in the revised MS.

**RC8.  *Leaching experiments of rock and surface soil samples with 0.5 M HCl do not account for the preferential Ca dissolution from carbonates as silicate dissolution is a much slower process.***

We did not intend to claim that the results from the leaching experiments account for the differences in reaction kinetics for carbonate vs silicate dissolution. As stated in Section 5.1 in the

original MS, the results from these experiments were used to assess the Si contents in the acid-soluble fraction of the bedrock. As stated in the original manuscript, the maximum potential contribution of Si from carbonate dissolution in fresh groundwaters can then be assessed using the Ca concentration of fresh groundwaters. It is possible that further Si could be released due to the effects of carbonate recrystallisation processes. However, the incorporation factor of Si into carbonates is known to be very low.

**RC9.   Fig 2a is redundant. Club spatial distributions of TDS and dissolved Si in Fig. 5**

Figure 2 shows the spatial distributions of a) TDS concentrations, b) dSi concentrations and c) $\delta^{30}$Si values as a function of latitude and depth. As there are no simple correlations between TDS and dSi, or TDS and $\delta^{30}$Si, we respectfully do not think that Fig 2a is redundant. Supporting text explaining this justification is now provided in the revised MS. Moreover, Figure 5 shows the spatial distributions of $\delta^{30}$Si values as a function of latitude and longitude at different depths (a: 0 to -1 m AHD, b: -1 to -5 m AHD, c: -5 to -15 m AHD), corresponding to the depths of groundwaters in each classification type (fresh, T1 or T2). It is not clear to the authors how it would be possible to "Club spatial distributions of TDS and dissolved Si" into this figure, but we are very open to improving the presentation of our figures.

**RC10. Fig. 3b: Low δ30Si in deep and saline groundwaters than expected from the seawater mixing (not shown) between 5-90 and seawater suggests in situ release of lithogenic Si (low δ30Si).**

This is an interesting observation by the reviewer that we had not duly considered. We agree that the lower $\delta^{30}$Si values of some saline groundwaters relative to conservative mixing suggests that lithogenic silica is released in the deeper aquifer. This is now discussed in the revised MS.

**RC11. Fig. 5 High and low δ30Si values are seen for groundwater collected near and away from the Salt lakes located on eastern side of the island. This hints at the probable mixing between a significant in situ release of lithogenic Si (low δ30Si) and low Si (high δ30Si) in seepage waters of the salt lakes. The salt lake waters and sediment porewaters should also be measured.**

The hypothesis that the salt lakes on Rottnest Island are hydrologically connected to the groundwater aquifer is at odds with the conclusions of previous studies on Rottnest Island. Algalcyanobacterial mats and mud sediments on the lake floors appear to act as aquitards that prevent interaction with groundwater from below (Playford, 1997). Although Bryan et al. (2016) observed some groundwater seepage from the freshwater lens around some lake edges, there was no geochemical evidence (major/trace element or stable isotope composition) for any significant influence of the lakes on the groundwaters.

A separate study focussing on the salt lakes on Rottnest Island currently ongoing and the data are not included in this study. However, we can confirm that the dSi concentrations of lakewaters sampled in the same field season as the majority of the groundwaters (September 2014) were all <35.7 µM. Thus, the dSi concentrations of the salt lakes are at least two-fold lower than the fresh and saline groundwaters. This justification for not considering the salt lakes as an important source of dSi is now provided in the revised MS, in addition to discussion regarding the potential diffusion of Si from the salt lakes.

***RC12. Fig. 6: It has been used to show older groundwaters contain high δ30Si due to more preferential removal of lighter isotopes on Fe-Al oxyhydroxides. However, the groundwater 25-90 has the lowest δ30Si (1.72) and much lower 3H (0.04) among older deep water. This misleading figure contradicts Fig. 3b indicating in situ release of lithogenic Si (low δ30Si).***

We thank the reviewer for highlighting this and we have modified our interpretation accordingly. We agree that lithogenic silica is released in the deeper aquifers. We now state that two competing processes appear to be occurring: 1) diffusion processes generating higher $\delta^{30}$Si values and 2) lithogenic silica dissolution releasing lower $\delta^{30}$Si silica. This is stated in the revised MS.

***RC13. Section 5.4: The additional source of Si (high δ30Si) from the salt lakes excluded in this study can potentially bias dissolved Si isotope composition for the marine SGD component.***

We outlined our justifications for why we consider that the groundwaters are not connected to the salt lakes in AR11, namely the lake sediments/algal mats acting as aquitards and the lack of geochemical evidence for lakewater input in groundwaters. Regarding the marine SGD component, the salt lakes likely represent a low concentration source of dSi and, therefore, we do not think that this input would bias dissolved Si isotope composition for the marine SGD component. In any case, marine-influenced hypersaline lakes are a common feature of other carbonate island aquifer systems, e.g. The Bahamas, and including this additional source of dSi in the marine SGD component would be correct.

***RC14. Table A2. Replace this table with Box plots of saturation indicies calculated for all samples and include Fe-Mn-Al minerals.***

The saturation indices were recalculated to include all Fe-Mn-Al minerals. However, many common Fe-Al-Mn phases (such as Ferrihydrite, Lepidocrocite, etc.) are not included in the phreeqc database, we originally used (phreeqc.dat). Thus, we have remodelled all saturation states using water4f.dat, which includes many of these minerals. Indeed, the results show that the groundwaters are saturated in many key Fe oxide minerals, such as goethite and Fe(OH)$_3$.

A box plot for all minerals will be included in the revised MS. However, producing a boxplot requires the selection of key minerals. To give an overview of all minerals, specifically the wide variety of possible Fe-Al-Mn phases, we have provided a table below showing their respective saturation indices (Table 2).

**Table 2 Example Phreeqc output for a fresh groundwater (RI_1-83)**

| Phase | Chemical formula | Saturation |
|---|---|---|
| Adularia | KAlSi3O8 | -2.91 |
| Al(OH)3(a) | Al(OH)3 | -2.81 |
| Albite | NaAlSi3O8 | -4 |
| AlumK | KAl(SO4)2:12H2O | -19.62 |
| Alunite | KAl3(SO4)2(OH)6 | -7.51 |
| Analcime | NaAlSi2O6:H2O | -5.23 |
| Anhydrite | CaSO4 | -2.26 |
| Annite | KFe3AlSi3O10(OH)2 | -0.93 |
| Anorthite | CaAl2Si2O8 | -6.6 |
| Aragonite | CaCO3 | -0.1 |
| Artinite | MgCO3:Mg(OH)2:3H2O | -6.39 |
| Basaluminite | Al4(OH)10SO4 | -7.01 |
| Beidellite | (NaKMg0.5)0.11Al2.33Si3.67O10(OH)2 | -2.34 |
| Birnessite | MnO2 | -11.64 |
| Bixbyite | Mn2O3 | -10.97 |
| Boehmite | AlOOH | -0.63 |
| Brucite | Mg(OH)2 | -5.34 |
| Calcite | CaCO3 | 0.05 |
| Chalcedony | SiO2 | -0.58 |
| Chlorite14A | Mg5Al2Si3O10(OH)8 | -6.81 |
| Chlorite7A | Mg5Al2Si3O10(OH)8 | -10.27 |
| Chrysotile | Mg3Si2O5(OH)4 | -5.51 |
| Clinoenstatite | MgSiO3 | -3.93 |
| CO2(g) | CO2 | -2.15 |
| Cristobalite | SiO2 | -0.53 |
| Diaspore | AlOOH | 1.15 |
| Diopside | CaMgSi2O6 | -5.03 |
| Dolomite | CaMg(CO3)2 | 0.25 |
| Dolomite(d) | CaMg(CO3)2 | -0.33 |
| Epsomite | MgSO4:7H2O | -4.26 |
| Fe(OH)2.7Cl.3 | Fe(OH)2.7Cl0.3 | 5.72 |
| Fe(OH)3(a) | Fe(OH)3 | 0.73 |
| Fe3(OH)8 | Fe3(OH)8 | -1.66 |
| Forsterite | Mg2SiO4 | -9.42 |
| Gibbsite | Al(OH)3 | -0.04 |
| Goethite | FeOOH | 6.31 |
| Greenalite | Fe3Si2O5(OH)4 | -7.31 |
| Gypsum | CaSO4:2H2O | -2.01 |
| H2(g) | H2 | -23.05 |
| H2O(g) | H2O | -1.74 |
| Halite | NaCl | -6.19 |
| Halloysite | Al2Si2O5(OH)4 | -4.71 |
| Hausmannite | Mn3O4 | -13.4 |
| Hematite | Fe2O3 | 14.59 |
| Huntite | CaMg3(CO3)4 | -3.72 |
| Hydromagnesite | Mg5(CO3)4(OH)2:4H2O | -13.28 |
| Illite | K0.6Mg0.25Al2.3Si3.5O10(OH)2 | -2.76 |
| Jarosite(ss) | (K0.77Na0.03H0.2)Fe3(SO4)2(OH)6 | -7.48 |
| Jarosite-K | KFe3(SO4)2(OH)6 | -8.1 |
| Jarosite-Na | NaFe3(SO4)2(OH)6 | -10.55 |
| JarositeH | (H3O)Fe3(SO4)2(OH)6 | -16.01 |
| Jurbanite | AlOHSO4 | -6.75 |
| Kaolinite | Al2Si2O5(OH)4 | 0.45 |
| Kmica | KAl3Si3O10(OH)2 | 2.57 |
| Laumontite | CaAl2Si4O12:4H2O | -3.22 |
| Leonhardite | Ca2Al4Si8O24:7H2O | 1.63 |
| Magadiite | NaSi7O13(OH)3:3H2O | -10.18 |
| Maghemite | Fe2O3 | 4.85 |
| Magnesite | MgCO3 | -0.35 |
| Magnetite | Fe3O4 | 13.75 |
| Manganite | MnOOH | -4.9 |
| Melanterite | FeSO4:7H2O | -8.9 |
| Mirabilite | Na2SO4:10H2O | -6.7 |
| Mn2(SO4)3 | Mn2(SO4)3 | -61.99 |
| MnCl2:4H2O | MnCl2:4H2O | -13.01 |
| MnSO4 | MnSO4 | -12.63 |

| | | | |
|---|---|---|---|
| Montmorillonit | -29.69 (HNaK)0.14Mg0.45Fe0.33Al1.47Si3.82O10(OH)2 | e-Aberdeen | |
| Montmorillonit | .58 -34.91 (HNaK)0.09Mg0.29Fe0.24Al1.57Si3.93O10(OH)2 | e-BelleFour | |
| Montmorillonit | 8 Ca0.165Al2.33Si3.67O10(OH)2 | e-Ca -2.28 | |
| Nahcolite | NaHCO3 | | -4.13 |
| Natron | Na2CO3:10H2O | | -8.36 |
| Nesquehonite | MgCO3:3H2O | | -2.75 |
| Nsutite | MnO2 | | -10.6 |
| O2(g) | O2 | | -40.05 |
| Phillipsite | Na0.5K0.5AlSi3O8:H2O | | -3.48 |
| Phlogopite | KMg3AlSi3O10(OH)2 | | -8.54 |
| Portlandite | Ca(OH)2 | | -11.52 |
| Prehnite | Ca2Al2Si3O10(OH)2 | | -6.93 |
| Pyrochroite | Mn(OH)2 | | -6.29 |
| Pyrolusite | MnO2 | | -10.81 |
| Pyrophyllite | Al2Si4O10(OH)2 | | 1.34 |
| Quartz | SiO2 | | -0.13 |
| Rhodochrosite | MnCO3 | | -0.32 |
| Rhodochrosite( | MnCO3 | d) -1.03 | |
| Sepiolite | Mg2Si3O7.5OH:3H2O | | -4.53 |
| Sepiolite(d) | Mg2Si3O7.5OH:3H2O | | -7.2 |
| Siderite | FeCO3 | | -2.17 |
| Siderite(d)(3) | FeCO3 | | -2.56 |
| Silicagel | SiO2 | | -1.12 |
| SiO2(a) | SiO2 | | -1.45 |
| Talc | Mg3Si4O10(OH)2 | | -3.09 |
| Thenardite | Na2SO4 | | -8.05 |
| Thermonatrite | Na2CO3:H2O | | -10.19 |
| Tremolite | Ca2Mg5Si8O22(OH)2 | | -8.24 |
| Trona | NaHCO3:Na2CO3:2H2O | | -14.36 |
| Wairakite | CaAl2Si4O12:2H2O | | -7.76 |

**Suggestions and editorial corrections:**

**RC15. Do not club multiple parenthesis.**

The instance of this we could find was a mistake on LNs 37-38 in the original MS. This is now amended in the revised MS.

**RC16. Conservative mixing is misleading due to in situ Si release and thus change it to theoretical mixing.**

"Conservative mixing" is now amended to "theoretical mixing" in the revised MS.

**RC17. Line 118: delete space after "stabilized"**

This erroneous space has been deleted in the revised MS.

**RC18. Line 131: add cm after "18.2 megaOhm"**

This has been updated in the revised MS to '18.2 MΩcm'

**RC19. Line 147: add Si after 28 and 29**

Si has been added after 28 and 29 so in the revised MS it now reads: '$^{28}$Si, $^{29}$Si and $^{30}$Si'.

**RC20. Line 285: Delete "and northwestern"**

This has been deleted in the revised MS

**RC21. Line 327 and elsewhere: change "rock-water" to "aquifer-water"**

"Rock-water" has been updated to "aquifer-water" in the modified manuscript where applicable, except in the introduction where we also discuss riverine samples and "aquifer-rock" would not be appropriate.

**REVIEWER #2**

*Review of manuscript HESS-2020-429 submitted to Hydrology and Earth System Sciences by Martin and colleagues: The evolution of stable silicon isotopes in a coastal carbonate aquifer, Rottnest Island, Western Australia*

*Martin et al present silicon isotope ratios (expressed as δ30Si) from fresh and saline groundwater samples from a carbonate island in Western Australia. They interpret variation in δ30Si of the freshwater samples to be driven by sorption of Si to Fe/Al oxides, while the saline water samples are governed by mixing with a low [Si], high d30si endmember. Given a recent focus on topics like e.g. boundary exchange, coastal filtering, SGD, etc. for the trace element and isotope budgets of the ocean, this paper has potential to be an interesting case study.*

*In general, the manuscript is well written and seems appropriately referenced. The topic is an interesting one, I think, though does seem to fall slightly outside the scope of HESS as I understand it. I have some methodological concerns (detailed below), and some comments about the interpretations, some more major than others. I list these below in order they appear in the manuscript.*

*Comments*

**RC22. L71: this suggestion of interaction with basement rocks is interesting but never returned to. If it can be relevant for Li, why not for Si?**

The reviewer makes an excellent point that we had overlooked and may explain some of the lower $\delta^{30}$Si values of the T2 groundwaters, relative to what would be expected for theoretical mixing of 5-90 (highest dSi groundwater) with seawater. This is now included in the revised manuscript discussion in Section 5.3.2.

**RC23. L95: relevance?**

In L95, we described how the recent decrease in rainfall and previous sea-level highstand events caused seawater intrusion into aquifer. The relevance of this was to highlight that intruded seawater may have adsorbed cations onto the aquifer matrix, which may bereleased into solution via cation exchange processes. This is now explicitly stated in Section 2.

**RC24. L100, 103: spell out units here – TU = tritium units? pMC = percent modern carbon? G PES = ??**

The units are now spelled out for TU and pMC. 'G PES' was a typing error and should have read 'g L$^{-1}$'. These are now updated in the revised manuscript.

**RC25. L104 vs. L98 – repetition, but with age expectations reversed. Not sure what's going on here.**

This was a mistake whereby the second instance on L104 should have read "T2 groundwaters" rather than "fresh groundwaters".

**RC26. L127: Some indication of precision and accuracy needed here.**

The accuracy of cations and anions measurements were evaluated the charge balance error, with 80% of the samples falling within ± 5% and all samples falling within ± 6.2%, which is much greater than the uncertainity than the analytical precision of measurements. This is now included in the revised MS.

**RC27. L132: This column protocol works sufficiently well for samples with low anionic components in the matrix. But for the saline samples, anions like sulphate or chlorine would elute at the same time as the sample. This could cause matrix effects between brack- eting NBS28 standards and samples. Previous work has shown that matrix effects, sensu lato, can induce large bias to silicon isotope ratios (e.g. Hughes et al. 2011 JAAS. What steps were taken to correct for this, or demonstrate that it is not a problem? Typically, a pre-concentration step like the so-called 'MAGIC' protocol is used for brackish and marine samples, since this also has the effect of removing much of the matrix.**

Unfortunately, the authors were not aware of this work by Hughes et al., (2011, JAAS) at the time the work was conducted and did not dope the standards with $H_2SO_4$ for MC-ICPMS measurements. However, we now clearly state this limitation of our measurements in the revised manuscript to highlight this for the reader.

We were aware of the brucite co-precipitation method for saline samples and tested this on the salt lake samples, but did not obtain 100% Si yields for these samples. This was further made difficult by the low Si concentrations in seawater and salt lake samples from Rottnest Island, which were below the limit of detection for Si at our facilities (3.6 µM).

***RC28. L137: Drying to 'incipient' dryness is also slightly worrying. What does this mean, in practice? If silica precipitates because it becomes oversaturated in the solution, there is a danger that it will not redissolve in 2% HNO3. And because the precipitation is likely associated with a fractionation, this may also induce bias. Did the authors demonstrate 100% Si yield at this stage of protocol?***

To obtain sufficiently high concentrations for Si isotopic measurements, and not overload the capacity of the resin by loading a high volume of sample, we gently evaporated the solutions at 80degC until a small wet blob of solution remained. Care was taken not to completely evaporate the solution and form precipitates. No precipitates were observed for the redissolved samples. Our column yields following this method were 100% (within analytical error).

***RC29. L140 or around: Give details of procedural blanks, for column chemistry and for alkali fusion***

As the limit of detection for Si at our facilities was relatively high (3.6 µM), we could not quantify the Si concentrations directly. Instead, we compared the signal intensities of our total procedural blanks for column chemistry (including blanks from alkali fusion that were processed through column chemistry)  with our acid blanks by MC-ICPMS as a screening process and did not detect any measureable Si signal.

We now state in the revised MS that the $^{30}$Si signal intensities for total procedural blanks could not be distinguished for the background measured in 2% (v/v) HNO$_3$, typically <130 mV compared to standard and sample intensities of ~25 V. Thus, the blank contribution was less than 0.3% and all Si loaded onto the columns was eluted in the Si fraction.

***RC30. L147: Give 'Si' for 28 and 29.***

These are now stated as '$^{28}$Si, $^{29}$Si' in the revised manuscript.

***RC31. L148: M/ΔM = 2000 is probably at the low end of resolution with which the polyatomic interferences can be avoided, and could induce some noise to the measurement if some of the interference peaks overlap onto the measurement plateau (though the three-isotope plot shows this isn't a large problem – uncertainties should be given in this plot).***

As stated by the reviewer, we believe that the plot of 30Si/28Si vs 29Si/28Si supports the accuracy of our measurements. The uncertainties for measurements are now provided on the plot and this is also provided below.

[Figure]

**Figure A.1 revised**

***RC32. L155: This implies only 1 bracket was measured – is this correct? Normally, 3-5 individual standard-sample brackets are analysed per sample. If this is correct, the stated ±0.12 uncertainty may be too precise.***

As already stated in our response to RC2, it would have been preferable to measure samples in triplicate, but unfortunately resources allocated for analyses in the project were limited by budget and time constraints. Following the suggestion of reviewers 1 and 2, we now include the internal uncertainty (2 standard error; 2SE) of the $\delta^{30}Si$ measurement for each sample in Table 1 of the revised MS and this is provided above in the response to RC2.

***RC33. L158: There are more recent values for IRMM-018 – see e.g. Geilert et al. 2020 (Nat Comms); Baronas et al. 2018 (EPSL), etc.***

We thank the reviewer for bringing this to our attention. The $\delta^{30}Si$ values for IRMM-018a from Geilert et al., (2020) and Baronas et al. (2018) were $-1.46 \pm 0.09$‰ and $1.57 \pm 0.08$‰, which are similar to our measured values. This is now stated in the revised manuscript.

***RC34. L161: Not sure the uncertainties of ±0.000 are required here.***

These uncertainties have been deleted in the revised MS.

***RC35. Figure 2: Panel A and B – the colorbar scale seems to imply the concentrations are negative.***

We understand the confusion due to the distance between the tick marks and the numbers. This has been increased in the revised figure for clarity and is shown below.

[Figure]

**Figure 2 revised**

***RC36. Figure 2: It's not clear which data are included in this. There is a red line in Fig 1, but presumably the samples falling off this line have been incorporated somehow. More detail could be useful.***

We apologise for the lack of clarity regarding which data was included in the plot. In the revised MS, we now state that all Rottnest Island groundwaters that we analysed were included to create these plots. To show the plots in two dimensions, we reduced sample locations onto a central axis (same longtitude) by interpolation and then created a grid based on their longitude and depth. This is now clearly stated in the revised manuscript caption.

***RC37. L172: I find the conclusion that T1 DSi is higher than fresh and T2 very hard to believe given Fig. 3A.***

We have double checked our calculations and got the same results. We also wanted to be sure originally and hence wanted to confirm that the difference was statistically valid, which is confirmed by the p values (0.001 and 0.002) being below our significance threshold of 0.05. On Figure 3A, you can also see there are two T1 groundwaters with much higher dSi concentrations than the fresh groundwaters and also the majority of the T1 groundwaters plot above the T2 groundwaters, excluding 5-90.

***RC38. I am also skeptical about the correlation on L171.***

Again, we have double checked our calculations to confirm our result. This p value (0.04) is clearly close to our defined significance threshold of 0.05, but nevertheless below it and should be regarded as statistically valid in this context.

***RC39. L191: extra ")"***

We thank the reviewer for highlighting this typing error and it is now corrected in the revised MS.

***RC40. Results section: I can't see the Si concentration and δ30Si for the seawater sample mentioned on L114. In e.g. Fig 4B, a literature value is used from Singh et al., but this is not sufficiently justified.***

The data for the seawater sample is now included in Table 1 in the revised MS. Unfortunately, due to the very low dSi concentration of local seawater at Rottnest Island, the $\delta^{30}$Si was not measured as a preconcentration procedure to remove the seawater matrix, such as MAGIC, was not established at our facilities.

The value for $\delta^{30}$Si from Singh et al. (2015) was included as it was the only available data from the Indian Ocean. We now state that although the Bay of Bengal likely does not represent the local seawater around Rottnest Island, the inverse relationship between dSi and $\delta^{30}$Si in seawater is well-established and has been explained by the preferential incorporation of lighter Si isotopes by diatoms (Grasse et al., 2013; Singh et al., 2015).

***RC41. L262: I disagree that there is a threshold of Al concentration necessary for Si isotopes to fractionate – pure SiO2 experiments also show fractionation (as noted elsewhere in this manuscript).***

We agree that we may have focussed too much on the results from Oelze et al. that only found Si isotopic fractionation in the presence of Al and not properly considered findings from previous studies, e.g. Geilert et al. 2014 (GCA). We have now removed this statement in the revised manuscript.

***RC42. L264: If the interpretation for d7Li is the same as for δ30Si, why is there no relationship between the two? (L189)***

We thank the reviewer for pointing out the inconsistency in our reasoning. In the revised MS, we now highlight the lack of correlation between $\delta^{7}$Li and $\delta^{30}$Si in RI groundwaters, and present the following hypothesis to explain this, namely the contrast between high concentrations of Li in seawater relative to fresh meteoric groundwaters, whereas dSi is a nutrient and depleted in seawater. This is an important distinction on RI since modern seawater intrusion (Bryan et al., 2016), and past sea-level high stands (~2 m higher than present), e.g., events at ~4 and 7 ka (Coshell and Rosen, 1994; and Gouramanis et al., 2012), would have probably intruded seawater into the shallow groundwater system. Such seawater intrusion episodes would be expected to

adsorb Li, but not Si, onto the aquifer matrix during previous/ongoing and provide Li to groundwaters through cation exchange processes.

**RC43. L265: the evidence for Fe oxides playing an important role seems to be circumstantial. Is there any direct evidence they are present, with sufficient sorption capacity, to be important?**

As outlined in our response to RC14, we have recalculated our saturation indices using phreeqc. This shows show that the groundwaters are saturated in many key Fe oxide minerals, such as goethite and $Fe(OH)_3$. A box plot for all minerals will be included in the revised MS and included in the discussion.

**RC44. Section 5.2: There is no real explanation for the positive trend between Si concentration and isotope ratios – this is a bit counter-intuitive. One might expect a negative correlation between Si concentration and isotope ratio.**

Reviewer 1 in RC6 presented an interesting point about the potential role of diffusion in modulating the groundwater $\delta^{30}Si$ values of fresh on Rottnest Island. We have attempted to further clarify this point in the revised MS. We had already attempted to discuss the role of physical mixing and diffusion in Section 5.2 in LNs 284-297. We have attempted to further clarify this point in the revised MS.

**RC45. L305: missing word (divided?)**

We thank the reviewer for pointing out this mistake and have corrected it in the revised MS.

**RC46. L312: See also above – this does not appear to be the case from Fig 3A.**

The reviewer here is referring to our statement that the T1 groundwaters have significantly higher concentrations than the fresh and T2 groundwaters. As we outlined in RC38, and stated in the original MS, this statement is supported by statistical tests that show the p values are less than our threshold for statistical significance (0.05)

**RC47. Section 5.3.1. I struggle to follow the rationale for including 'T1' groundwaters as a separate case rather than just viewing them as intermediate between freshwaters and the marine/T2 samples.**

We agree with the reviewer and have removed "T1 groundwaters" from the title of Section 5.3.1 in the revised MS.

*RC48. L335 and around – the challenge seems to be to define the non-marine endmember. If this was more rigorously achieved, then the discussion might be more usefully structured in terms of deviations from conservative mixing as in e.g. estuary papers (Zhang et al. 2020 GCA).*

We thank the reviewer for bringing this paper to our attention. We agree that having a well-defined fresh/non-marine end-member would be advantageous. However, it can be seen from Fig 3A and 3B in the original MS that dSi concentrations and Si isotopic ratios do not show a simple mixing relationship with Cl concentrations (and by extension, salinity). Moreover, the dSi concentrations and Si isotopic ratios of the fresh groundwaters were highly variable, despite have very low salinities. This might reflect the fact that RI has a small carbonate island aquifer, which is not comparable to a large river, such as the Amazon or Yangtze, flowing into an ocean or sea.

*RC49. In general, it's also not clear where the marine endmember comes from. I don't think a value from the Bay of Bengal is necessarily very representative for waters bathing western Australia (see e.g. Holzer et al. 2015 GBC)*

As stated in our response to RC40, we now state in the revised MS that although the Bay of Bengal likely does not represent the local seawater around Rottnest Island, but may be realistic given the low dSi concentration of local seawater. This is due to the inverse relationship between dSi and $\delta^{30}Si$ in seawater is well-established and has been explained by the preferential incorporation of lighter Si isotopes by diatoms (Grasse et al., 2013; Singh et al., 2015).

*RC50. L355: Could/should this not instead yield low δ30Si values if there is a re-equilibration between solute and rock?*

We agree with the reviewer and now state in the revised MS that low $\delta^{30}Si$ in the deeper aquifer may result from the equilibration processes between the groundwaters and the rock.

*RC51. L372: I don't understand this – wouldn't these values be representative of the terrestrial component of SGD?*

We agree with the reviewer and this is now considered to represent the terrestrial SGD component , i.e., 'fresh SGD'.

**RC52. It is disappointing there is no attempt to calculate fluxes and place them in a global context. In general, the global implications section is rather weak and seems added on as an afterthought. It would be good to see a better discussion of how the lessons from this case study can (or cannot) be applied elsewhere, for example, and a more rigorous attempt at upscaling.**

We agree with the reviewer and now estimate this in the revised manuscript. The terrestrial SGD flux from Western Australia is estimated to be 3.2 $km^3/a$ with carbonate aquifers comprising 50% of the coastline (Zekster et al. 2007; Rahman et al., 2019). By adopting the average dSi concentration of the fresh groundwaters (112±23 μM), we estimate that the dSi flux from the Tamala Limestone is 1.79 $x10^8$ ± 0.37 $x10^8$ mol/a of dSi, corresponding to ~27% of the dSi flux from Western Australia.

**REVIEWER #3**

**RC53. The manuscript is well written, providing a critical dataset in an understudied system. I enjoyed reading it. The figures are generally clear. I have suggested some minor revisions and have some suggestions for the discussion. There were some few grammatical errors. I don't anticipate the revisions would take the authors very long to incorporate.**

We thank the reviewer for their kind comments and have addressed their individual comments on a point-by-point basis in the following section.

**RC54. Suggested changes throughout the paper: I prefer "bSi" to "BSi" as the capital "B" in "BSi" can be mistaken for the symbol for Boron. Similarly, I suggest changing "DSi" to "dSi" so that the capital D won't be mistaken for Deuterium. For readers versed in the biogenic silica literature, the capital letters may not pose a problem. However, for new readers it may contribute to some confusion.**

We had not considered that this may be misleading, but agree with the reviewer's suggestion. This is now changed in both this document and the revised MS.

*RC55. Well written introduction. Clear reasoning, concise review, and compelling set-up and motivation for the project to fill a known data gap. I suggest adding Ehlert et al, 2016 (Ehlert, C., Reckhardt, A., Greskowiak, J., Liguori, B.T., Böning, P., Paffrath, R., Brumsack, H.J. and Pahnke, K., 2016. Transformation of silicon in a sandy beach ecosystem: Insights from stable silicon isotopes from fresh and saline groundwaters. Chemical Geology, 440, pp.207-218) to the 2nd paragraph (lines 42-54). That data was not included in Frings et al 2016.*

We thank the reviewer for highlighting that this data was not included in Frings et al. 2016 and agree that it is an important reference. We have now included this citation in the introduction.

*RC56. 2. Study Area The last paragraph is confusing w.r.t. description of "fresh" mixing type. One set of parameters is given for this zone in lines 99-101, and another set is given later in the paragraph (lines 103-105), with some repeated conditions (e.g., above 1m AHD).*

This was a mistake whereby the second instance on L104 should have read "T2 groundwaters" rather than "fresh groundwaters". This was already addressed in response to RC25 and is now correctly written in the revised MS.

*RC57. Also would it be possible to add a salinity or [Cl-] range to the three mixing zones?*

We thank the reviewer for their suggestion. This will be added to the figure in the revised MS.

*RC58. 3. Methods: No salt corrections or formation of a brucite precipitate was conducted to account for matric effects prior to purification on via Biorad AG 50WX8?*

We only processed samples that had sufficiently high dSi and low TDS so that they could be processed without exceeding the resin capacity, typically <10%, and a brucite coprecipitation was not necessary.

*RC59. 4.Results Do the authors have a surface coastal water/open ocean value for dSi? I think the manuscript would be improved by adding a comparison of d30Si found in fresh-T2 to Ehlert et al 2016 in the second paragraph (lines 176-186).*

The value for $\delta^{30}Si$ from Singh et al. (2015) was included as it was the only available data from the Indian Ocean. We now state that although the Bay of Bengal likely does not represent the local seawater around Rottnest Island, the inverse relationship between dSi and $\delta^{30}Si$ in seawater is well-established and has been explained by the preferential incorporation of lighter Si isotopes

by diatoms (Grasse et al., 2013; Singh et al., 2015). We have also included the data from Ehlert et al. to be added to the revised MS.

**RC60. Incidentally, have the authors seen Mayfield et al., 2021 (Nature Communications)?**

The authors were aware of this paper as we actually provided some samples from Rottnest Island that form part of the dataset. As this paper has been published since we submitted, we will appropriately discuss it in the revised MS.

**RC61. Line 190: missing punctuation at end of sentence Line 191: extraneous parenthetical after "respectively"**

We thank the reviewer for pointing this out and it is now corrected in the revised manuscript.

**RC62. Fig. 2: So cool! Regarding the legend for TDS and dSi: are those negative values?**

We understand the confusion due to the distance between the tick marks and the numbers. This distance has been increased in the revised figure for clarity. The revised figure is given in our response to RC35.

**RC63. Fig. 4: Can authors add a regression line to 4a as this is discussed in results and discussion?**

A regression line has been added to Figure 4a in the revised MS.

**RC64. Also, in 4b, there are 2 green dashed lines and only description of 1 green dashed line in the caption.**

We thank the reviewer for noticing this and have updated the caption for Figure 4b in the revised MS accordingly.

**RC65. 5.1 Evidence for the source of DSi. . . Lines 223-226: referring to a figure here would be helpful**

This is a good suggestion and a reference to Figure 4a has been provided in the revised MS.

*RC66. Would you expect lower d30Si if RI was behaving like a closed system which could be described via Rayleigh distillation model? To clarify, if almost all the dSi supplied by quartz dissolution formed a secondary clay mineral, would the subsequent dissolution of that secondary mineral would impart a dissolved d30Si signature more like the primary mineral? This scenario could be added to the discussion of secondary mineral dissolution in lines 230-235), unless the authors have other evidence this is not likely.*

The reviewer raises an interesting and important point that should be expanded on in the revised MS. It might be expected if most of the dSi supplied by quartz dissolution formed a secondary clay mineral, which was subsequently dissolved ,this would produce a dissolved d30Si signature that was lower than the primary mineral. This has be added to the discussion of secondary mineral dissolution in the revised MS.

*RC67. Line 237: regarding bSi solubility, coastal settings can actually make for recalcitrant diatom frustules: proximity to dissolved Al and metals can help preserve frustules (e.g., solubility studies by Van Cappellen, Van Bennekom, and Loucaides et al 2012). I suggest authors clarify this sentence with specific conditions of their study site or leave out the last phrase about preservation in coastal sediments.*

In light of the reviewer sharing their expertise and what has been found in the relevant published works, our statement on the preservation of diatoms in coastal sediments has been removed.

*RC68. Could the authors add some site specific data w.r.t. to dissolved Fe concentrations in their discussion of adsorption being the likely mechanism that takes up light isotope of dSi.*

As outlined in our response to RC4, we now include these Fe data in the revised MS in Table 1. The revised version of Table 1 is also provided in our response to RC2.

*RC69. What were the concentrations of Fe in the 0.5M HCl leachate?*

The data from the leaching experiments is provided below (Martin et al., 2020). As readers may not have access to this publication, we will provide this in the supplementary information of the revised MS.

| Sample | Li | Sr | Ca | Al/Ca | Fe/Ca | K/Ca | Li/Ca | Mg/Ca | Na/Ca | Si/Ca | Sr/Ca | Zr/Ca | Sr/Li |
|--------|----|----|----|-------|-------|------|-------|-------|-------|-------|-------|-------|-------|

| | | uM | uM | mM | uM/m | uM/m | uM/m | uM/m | uM/m | uM/m | uM/m | uM/m | uM/m | uM/u |
|---|---|---|---|---|---|---|---|---|---|---|---|---|---|---|
| B01 | Acid soluble | 0.01 | 0.54 | 0.22 | 0.46 | 0.23 | 1.89 | 0.07 | 71.90 | 0.20 | 0.11 | 2.45 | 0.10 | 36.03 |
| S01 | Acid soluble | 0.01 | 0.55 | 0.22 | 0.62 | 0.15 | 0.29 | 0.06 | 79.27 | 0.20 | 0.05 | 2.48 | 0.05 | 41.70 |
| B01 | Residue | 0.52 | 16.35 | 3.90 | 5.01 | 5.19 | 2.58 | 0.13 | 115.70 | 1.44 | 4.95 | 4.20 | 2.90 | 31.40 |
| S01 | Residue | 0.33 | 10.13 | 2.78 | 12.84 | 3.11 | 9.04 | 0.12 | 124.01 | 2.02 | 10.75 | 3.65 | 7.64 | 30.28 |
| B01 | Bulk | 0.48 | 12.77 | 3.38 | 3.44 | 3.69 | 2.80 | 0.14 | 109.15 | 1.66 | 0.68 | 3.78 | 6.28 | 26.42 |
| S01 | Bulk | 0.35 | 13.61 | 3.54 | 5.10 | 1.29 | 4.03 | 0.10 | 122.92 | 1.58 | 3.37 | 3.84 | 3.99 | 38.52 |
| Av. | Acid soluble | 0.01 | 0.54 | 0.22 | 0.54 | 0.19 | 1.09 | 0.06 | 75.58 | 0.20 | 0.08 | 2.47 | 0.07 | 38.8 |
| Av. | Residue | 0.42 | 13.19 | 3.46 | 4.27 | 2.49 | 3.42 | 0.12 | 116.03 | 1.62 | 2.03 | 3.81 | 5.14 | 32.47 |
| Av. | Bulk | 0.18 | 7.08 | 1.88 | 2.86 | 0.72 | 2.16 | 0.08 | 101.1 | 0.89 | 1.71 | 3.16 | 2.02 | 40.11 |

[a]1SD uncertainty for all elements is ±6.2%.

**RC70. Are there dissolved Fe or O2 profiles at the sites? Are these sediments oxic?**

As outlined in our response to RC4, we now include these Fe and DO data in the revised MS in Table 1. The revised version of Table 1 is also provided in our response to RC2. These data show that the shallow groundwaters are generally oxic but become sub-oxic in the deeper T2 groundwaters.

**RC71. The more positive d30Si of dSi observed in groundwater in Ehlert et al 2016 was due to secondary mineral formation. Can the authors provide more evidence of why that's not occurring here? PHREEQC doesn't encompass the solubilities of all the amorphous-type alumino-silicates that can form.**

As outlined in our responses to RC14 and RC43, we have recalculated our saturation indices using phreeqc. This shows show that the groundwaters are saturated in many key Fe oxide minerals, such as goethite and $Fe(OH)_3$. The saturation indices were recalculated to include all Fe-Mn-Al minerals. As the reviewer highlights, many common Fe-Al-Mn phases (such as Ferrihydrite, Lepidocrocite, etc.) are not included in the phreeqc database that we originally used (phreeqc.dat). Thus, we have remodelled all saturation states using water4f.dat, which includes many of these minerals. Indeed, the results show that the groundwaters are saturated in many key Fe oxide minerals, such as goethite and $Fe(OH)_3$. These are provided in Table 2 of this document.

***RC72. Line 286: refer to a figure Line 305: missing a word or phrase here "...can be into two water types..."***

We thank the reviewer for pointing out this mistake. "Divided" has been added after "can be" in the revised MS.

***RC73. Line 325-328: This three end-member mixing concept is important: could it be highlighted in one of the figures and accompanied by a reference to that figure at the end of this sentence? Or is this in Figure 4b?***

We did attempt to show this to some degree in Figure 4B. In the revised manuscript, a three end-member mixing plot has been added to Figure 4.

***RC74. Conclusions highlight broad impacts, well written.***

We thank the reviewer for their kind feedback.

**References**

Bryan, E., Meredith, K.T., Baker, A., Andersen, M.S., Post, V.E. and Treble, P.C. (2020) How water isotopes (18O, 2H, 3H) within an island freshwater lens respond to changes in rainfall. Water research 170, 115301.

Grasse, P., Ehlert, C. and Frank, M. (2013) The influence of water mass mixing on the dissolved Si isotope composition in the Eastern Equatorial Pacific. Earth and Planetary Science Letters 380, 60-71.

Martin, A.N., Meredith, K., Norman, M.D., Bryan, E. and Baker, A. (2020) Lithium and strontium isotope dynamics in a carbonate island aquifer, Rottnest Island, Western Australia. Science of The Total Environment 715, 136906.

Playford, P.E. (1997) Geology and hydrogeology of Rottnest island, Western Australia.

Singh, S.P., Singh, S.K., Bhushan, R. and Rai, V.K. (2015) Dissolved silicon and its isotopes in the water column of the Bay of Bengal: Internal cycling versus lateral transport. Geochimica et Cosmochimica Acta 151, 172-191.

---

## Author Response (AR1)

**Author Response to Reviewers**

17 May 2021

Dear Editor,

We would once again like thank the three anonymous reviewers for taking the time to provide feedback. We have significantly revised the manuscript and believe that their feedback has significantly improved the quality of the manuscript. Specifically, we have fine-tuned our interpretations following insights from the reviewers, clarified details regarding analytical procedures and produced new figures, modifying existing figures where appropriate.

Our responses and associated edits resulting from these comments are given in this document. Please find the reviewer comments (RC) below in ***bold italics*** and our author responses (AR) in plain text. Where necessary, we have quoted the appropriate line numbers (LN) in the original or revised manuscript (MS). Note that at the request of Reviewer 3, we have changed "BSi" to "bSi" as the capital "B" in "BSi" can be mistaken for the symbol for boron. Similarly, "DSi" has been changed to "dSi" so that the capital "D" won't be mistaken for deuterium. This notation is also adopted in this response.

Best regards,

*Ashley Martin*

Dr. Ashley N. Martin (on behalf of all co-authors)

**RC1.** **This manuscript attempts to explore Si isotope dynamics of groundwater collected from Rottnest Island, Australia to quantify dissolved Si flux and its isotope composition ($\delta 30Si$) in the marine SGD component from coastal limestone aquifers. Authors claim that Si released by dissolution of lithogenic materials (paleosols) followed by its adsorption on Fe-Al (oxyhydroxides) is the key mechanism fractionating Si isotopes in shallow and relatively fresh groundwaters, while deep samples are affected by seawater mixing. Thus, Si isotope compositions of the sampled groundwater reflect the degree of aquifer-water interaction depending on groundwater residence time and local hydrogeology. The dataset has major analytical and interpretation issues.**

The reviewer has kindly summarised the findings in our MS, but unfortunately feels the dataset has major analytical and interpretation issues. We have addressed these concerns by responding to the specific comments outlined in this review and, where necessary, making the appropriate changes to the revised MS. We feel that these revisions have substantially improved the MS and would like to thank the reviewer for providing their insights.

**RC2.** **Accuracy and precision of Si isotope measurements are ensured from $\delta 30Si$ of -1.52 ± 0.12 per mil determined for IRMM-018a (n=5). However, for accurate Si isotope results, each sample is analyzed generally a minimum of three times as blank-standard-blank-sample-blank-standard and the average $\delta 30Si$ composition along with uncertainty (2s) is reported, which is not the case here. Given a single measurement for each sample, higher uncertainty of the order of 0.3-0.5 per mil is expected.**

As the reviewer stated, the accuracy and precision of our analyses were evaluated by repeat measurements of the reference material IRMM-018a, which were within error of reference values. We fully agree that it would have been preferable to measure samples in triplicate. However, resources allocated for analyses in the project were unfortunately limited by budget and time constraints. Following the reviewer's suggestion, we now include the internal uncertainty (2 standard error; 2SE) of the $\delta^{30}Si$ measurement for each sample in Table 1 of the revised MS and this is provided below. We apologise that these uncertainties were not included in the original submission as this was an oversight. The average 2 s.e. uncertainty of all sample measurements was 0.39‰, which is in line with the reviewers' estimated uncertainty of 0.3-0.5‰ for our analytical protocol. Although a lower level of uncertainty would be preferable, an uncertainty of ca. 0.4‰ does not affect our interpretations due to the large range of silicon isotopic ratios measured in groundwaters at Rottnest Island (-0.39 to +3.60‰), which is an order of magnitude higher than the uncertainty (ca. 4 vs 0.4‰). To better reflect this level of

uncertainty, we now report Si isotopic data to only 1 decimal place in the revised MS. The error bars in all relevant figures in the revised MS have been updated to reflect the 2 s.e. uncertainties.

**Table 1 (revised)**

| ID (RI-) | Water type[a] | Sampling date | Screen (m AHD) | Cl (mM) | DO (mg/L) | Al (uM) | Mn (uM) | Fe (uM) | dSi[b] (uM) | δ30Si (‰) | | 2 s.e. |
|---|---|---|---|---|---|---|---|---|---|---|---|---|
| 2-77 | F | 9/29/14 | -0.11 | 4.6 | 1.8 | <0.4 | <18 | <0.1 | 81.9 | +0.7 | ± | 0.4 |
| 3-77 | F | 9/29/14 | -0.01 | 5.1 | 1.5 | <0.4 | <18 | <0.1 | 89.0 | +0.7 | ± | 0.3 |
| 1-83 | F | 9/28/14 | 0.09 | 6.1 | 1.6 | <0.4 | <18 | <0.1 | 124.6 | +1.3 | ± | 0.5 |
| 4-83 | F | 9/30/14 | -0.11 | 4.4 | 4.2 | <0.4 | <18 | <0.1 | 113.9 | +1.0 | ± | 0.5 |
| 2-90 | F | 9/29/14 | -0.50 | 4.2 | 3.6 | <0.4 | <18 | <0.1 | 74.8 | +1.2 | ± | 0.5 |
| 6-90 | F | 9/29/14 | -0.64 | 6.8 | 3.4 | <0.4 | <18 | <0.1 | 135.3 | +2.3 | ± | 0.3 |
| 16-90 | F | 9/28/14 | -0.28 | 5.9 | 1.3 | <0.4 | <18 | <0.1 | 138.9 | +1.5 | ± | 0.6 |
| 17-90 | F | 9/30/14 | 0.06 | 5.9 | 3.5 | <0.4 | <18 | <0.1 | 121.1 | +1.1 | ± | 0.3 |
| 3-93 | F | 9/29/14 | -0.27 | 7.5 | 2.9 | <0.4 | <18 | <0.1 | 117.5 | +2.6 | ± | 0.3 |
| 6-93 | F | 9/29/14 | -0.22 | 5.5 | 4 | <0.4 | <18 | <0.1 | 131.7 | +1.6 | ± | 0.3 |
| 1-94 | F | 9/28/14 | -0.53 | 4.6 | 4.6 | 1.5 | <18 | <0.1 | 85.5 | +0.2 | ± | 0.4 |
| 2-94 | F | 9/28/14 | -1.00 | 2.7 | 7.1 | <0.4 | <18 | <0.1 | 128.2 | +1.0 | ± | 0.4 |
| 1-90 | T1 | 9/29/14 | -0.90 | 35.1 | 2.5 | <0.4 | <18 | <0.1 | 128.2 | +1.5 | ± | 0.5 |
| 8-90 | T1 | 9/29/14 | -0.59 | 9.2 | 1.9 | <0.4 | <18 | <0.1 | 192.3 | +2.8 | ± | 0.3 |
| 13-90 | T1 | 3/11/15 | -3.55 | 74.1 | 0.5 | <0.4 | <18 | <0.1 | NA | +1.2 | ± | 0.4 |
| 21-90 | T1 | 9/26/14 | -4.04 | 46.4 | 0.3 | <0.4 | 36 | 1.8 | 117.5 | +1.4 | ± | 0.3 |
| 24-90 | T1 | 9/26/14 | -3.47 | 11.9 | 0.4 | <0.4 | 91 | 2.0 | 131.7 | +1.2 | ± | 0.4 |
| 28-90 | T1 | 9/26/14 | -1.52 | 27.4 | 1.1 | <0.4 | <18 | <0.1 | 113.9 | -0.4 | ± | 0.3 |
| 3-94 | T1 | 9/27/14 | -0.72 | 13.7 | 3 | <0.4 | <18 | <0.1 | 78.3 | +1.8 | ± | 0.2 |
| 4-94 | T1 | 9/27/14 | -1.83 | 8.9 | 2.4 | <0.4 | <18 | <0.1 | 174.5 | +1.2 | ± | 0.4 |
| 5-94 | T1 | 9/27/14 | -1.87 | 8.9 | 1.1 | <0.4 | <18 | <0.1 | 128.2 | +0.6 | ± | 0.3 |
| 5-90 | T2 | 9/27/14 | -6.90 | 97.0 | 0.4 | <1.9 | 36 | <0.9 | 195.8 | +2.6 | ± | 0.2 |
| 11-90 | T2 | 9/26/14 | -6.19 | 473.2 | 0.5 | <1.9 | 146 | 3.3 | 81.9 | +3.6 | ± | 0.2 |
| 15-90 | T2 | 9/26/14 | -14.92 | 381.0 | 0.2 | <1.9 | 328 | 3.0 | 106.8 | +2.7 | ± | 0.4 |
| 18-90 | T2 | 9/27/14 | -11.16 | 409.1 | 0.3 | <1.9 | 146 | <0.9 | 64.1 | +2.8 | ± | 0.5 |
| 25-90 | T2 | 9/27/14 | NA | 225.0 | 0.3 | 30.4 | 55 | <0.9 | 99.7 | +1.7 | ± | 0.5 |
| 27-90 | T2 | 9/26/14 | -4.98 | 308.9 | 0.9 | <1.9 | <18 | <0.9 | 113.9 | +2.3 | ± | 0.4 |
| 31-90 | T2 | 9/28/14 | -9.20 | 244.2 | 0.3 | <1.9 | 1 470 | <0.9 | 85.5 | +2.1 | ± | 0.5 |
| SW-2 | SW | 29/9/15 | NA | 516.2 | 13.1 | <0.1 | <1 | <0.1 | <3.6 | NA | | |

**RC3. Samples are collected in two different seasons (Sep. 2014 and Mar 2015) but there is no mention of seasonality on the measured data.**

We agree that the Si isotopic ratios of samples may vary seasonally. However, the majority of the samples (27 out of 29) were collected in September 2014 and only two samples were collected in March 2015 to supplement the dataset (7-90 and 13-90). As only two samples were collected in March 2015 and do not contribute significantly to the overall dataset, we did not discuss the role of seasonality in the original MS. A key point to consider is that the mean residence times of fresh groundwaters, for which lumped-parameter modelling was conducted by Bryan et al.

(2020), is around 40 years (see Table A.3 in the original MS). Therefore, any seasonality is likely to be dampened by the multi-annual groundwater residence times. As the groundwater residence time increases with depth, any effects of seasonality will be even less relevant for the deeper T1 and T2 groundwaters. To address the reviewer's concern, we have removed data for 7-90 from Table 1 as no dSi or $\delta^{30}$Si were measured for this sample. The Si isotopic data for 13-90 has been kept in the revised MS as this sample does not appear to represent an outlier in relation to the other groundwaters and does not influence the discussion. The sample collection dates were already presented in Table 1 of the MS. In addition, we now state that one sample was collected in a different field campaign in the revised MS. However, we believe that a further discussion of seasonality is not warranted due to the relatively long groundwater residence times.

***RC4. Neither the reduction potential nor DO/Fe-Mn redox pairs are shown to reject the dissolution of Fe-Mn-Al oxyhydroxides in groundwater.***

We agree with the reviewer that including these data would be useful for assessing the importance of Fe-Mn-Al oxyhydroxide dissolution in groundwaters. The DO data were actually already included in the supplementary information file of the original submission, but we understand that this may not have been clear enough to the reader. The DO data were not included in the main MS as they were previously described by Bryan et al. (2016) and the authors wanted to ensure the originality of data presented in this MS and not repeat previous publications. The location of these data in the supplementary information file is now clearly stated in the revised MS and are provided in Table 1. The DO concentrations of fresh, T1 and T2 groundwaters ranged from 1.3-7.1, 0.3-3.3, and 0.2-0.9, respectively. These data suggests that the aquifer may become sub-oxic at depth. As a detailed description of the Rottnest Island aquifer was already provided by Bryan et al. (2016), we had attempted to avoid purposefully characterising fundamental aquifer processes in this MS. As we now see the need for additional such descriptions, particularly with relation to redox processes, this is now discussed in the revised MS.

The groundwater Al, Mn and Fe concentrations were not included in the original MS as they were generally below the limit of detection (LOD) and difficult to interpret on this basis. However, we now acknowledge that these data are of interest and should nevertheless be included in the manuscript. The Fe, Mn and Al concentrations are now included in Table 1 (given in Table 1 above in response to RC2) and are presented in the results section in the revised MS.

In all fresh groundwaters except groundwater 1-94, the Al, Mn and Fe concentrations were below the LOD. Interestingly, 1-94 also had the highest Al concentration (1.5 µM) and the lowest $\delta^{30}Si$ value (+0.2‰) of all the fresh groundwaters, which is now stated in the revised MS. The higher Al concentration and lower $\delta^{30}Si$ value may suggest an increased role for silicate dissolution in the sample and this is now discussed in the revised MS.

In T1 groundwaters, the Al, Mn and Fe concentrations were below the LOD in all samples except 21-90 and 24-90, which had Mn concentrations of 36 and 91 µM, respectively, and Fe concentrations of 1.8 and 2.0 µM, respectively. However, the dSi and $\delta^{30}Si$ values of these groundwaters are similar to other T1 groundwaters and do not appear to exhibit distinctive signatures for Fe-Mn oxyhydroxide dissolution. This is now discussed in the revised MS.

The T2 groundwaters had much higher Mn concentrations up to 1,470 µM, but generally low Al and Fe concentrations. The high Mn concentration for T2 groundwater 31-90 (1,470 µM) greatly differs from the average groundwater Mn concentration (98±282 µM, 1 s.d.) and may be considered as an outlier. However, as groundwater 31-90 also had the lowest $^{13}C_{DIC}$ and $^{13}C_{DOC}$ values, the elevated Mn may be related to the redox cycling of Mn by microbial processes. There is no correlation between Mn concentrations and dSi or $\delta^{30}Si$ values for the T2 groundwaters This is now discussed in the revised MS. Groundwater 25-90 had the highest Al concentration (30 µM) and the lowest $\delta^{30}Si$ value (+1.7‰). This is now discussed in the revised MS. This may suggest an increased role for silicate dissolution in this deeper part of the deeper aquifer.

***RC5.   Labels are wrong in Fig. 1c (1-90 repeated) and Fig. 5. Labels as mentioned somewhere in the text do not match with Table 1.***

We thank for the reviewer for bringing to our attention the error in Fig 1c. This error appears to result from overlapping label issues in the GIS software not displaying the sample ID for 11-90 correctly. An updated version of Fig 1c is now provided in the revised MS.

The Sample IDs in Table 1 were listed in the format RI_xx-xx, but did not have the 'RI_' precursor in the text. In the revised MS the Sample IDs in Table 1 are not listed without the 'RI_' precursor for consistency with the main text (as given in Table 1 above in response to RC2).

***RC6.   Dissolved Si and δ30Si of fresh groundwaters show a significant positive correlation, which indicates a dominating physical mixing/diffusion control on the Si isotope budget also supported by the narrow spatial extent of groundwaters collected from vertical depths <1 m only dispersed over 1-2 km.***

The reviewer raises an interesting point about the potential role of diffusion in modulating the groundwater $\delta^{30}$Si values of fresh on Rottnest Island. This hypothesis requires the identification of a high dSi/high $\delta^{30}$Si end-member, which we identify to be the T1 groundwaters. We had already discussed the role of physical mixing and diffusion in the original MS in Section 5.2 in LNs 284-297. Specifically, we highlighted that fresh groundwaters may be undergoing vertical mixing with older, more $\delta^{30}$Si-evolved T1 groundwaters. We have attempted to further clarify this point in the revised MS in Lines 269-289.

***RC7.  Overlapping wide variations in dissolved Si and δ30Si of groundwater from intermediate depths further corroborates this idea.***

As outlined above in our previous response (AR6), it might be argued that we already raised this point in our discussion in relation to vertical mixing of fresh groundwaters with older, more $\delta^{30}$Si-evolved T1 groundwaters. To further argue this, we now further discuss this in the revised MS Lines 269-289.

***RC8.   Leaching experiments of rock and surface soil samples with 0.5 M HCl do not account for the preferential Ca dissolution from carbonates as silicate dissolution is a much slower process.***

We did not intend to claim that the results from the leaching experiments account for the differences in reaction kinetics for carbonate vs silicate dissolution. As stated in Section 5.1 in the original MS, the results from these experiments were used to assess the Si contents in the acid-soluble fraction of the bedrock. As also stated in the original manuscript, the maximum potential contribution of Si from carbonate dissolution in fresh groundwaters can then be assessed using the Ca concentration of fresh groundwaters. It is possible that further Si could be released due to the effects of carbonate recrystallisation processes. However, the incorporation factor of Si into carbonates is known to be very low. As this is a minor point in our discussion and has caused confusion, we have removed this text from the discussion since it does not appear to add value to the discussion. Rather, in the revised MS, it is now stated in the introduction that we do not consider carbonate weathering to be important for Si isotopes (Lines 63-65).

***RC9.   Fig 2a is redundant. Club spatial distributions of TDS and dissolved Si in Fig. 5***

Figure 2 shows the spatial distributions of a) TDS concentrations, b) dSi concentrations and c) $\delta^{30}Si$ values as a function of latitude and depth. As there are no simple correlations between TDS and dSi, or TDS and $\delta^{30}Si$, we respectfully disagree that Fig 2a is redundant. Supporting text explaining this justification is now provided in the revised MS. Moreover, Figure 5 shows the spatial distributions of $\delta^{30}Si$ values as a function of latitude and longitude at different depths (a: 0 to -1 m AHD, b: -1 to -5 m AHD, c: -5 to -15 m AHD), corresponding to the depths of groundwaters in each classification type (fresh, T1 or T2). It is not clear to the authors how it would be possible to "Club spatial distributions of TDS and dissolved Si" into this figure, but we are very open to improving the presentation of our figures.

***RC10. Fig. 3b: Low δ30Si in deep and saline groundwaters than expected from the seawater mixing (not shown) between 5-90 and seawater suggests in situ release of lithogenic Si (low δ30Si).***

This is an interesting observation by the reviewer that we had not duly considered. We agree that the lower $\delta^{30}Si$ values of some saline groundwaters relative to conservative mixing suggests that lithogenic silica is released in the deeper aquifer. This is now discussed in the revised MS in Section 5.3 in terms of a three-component mixing model (Figure 4B).

***RC11. Fig. 5 High and low δ30Si values are seen for groundwater collected near and away from the Salt lakes located on eastern side of the island. This hints at the probable mixing between a significant in situ release of lithogenic Si (low δ30Si) and low Si (high δ30Si) in seepage waters of the salt lakes. The salt lake waters and sediment porewaters should also be measured.***

The reviewer raises an interesting point but we believe that hypothesis requires the identification of a high dSi/high $\delta^{30}Si$ end-member, but the dSi concentrations of these waterbodies are very low (<3.6 µM), which is an order of magnitude lower than the fresh groundwaters. This suggests that dSi is sourced from within the aquifer, which we identify to be the dissolution of lithogenic silica. Another important point is that the salt lakes on Rottnest Island are not considered to be hydrologically connected to the groundwaters. This is because algal-cyanobacterial mats and mud sediments on the lake floors appear to act as aquitards that prevent interaction with groundwater from below (Playford, 1997). Although Bryan et al. (2016) observed some groundwater seepage from the freshwater lens around some lake edges, there was no geochemical evidence (major/trace element or stable isotope composition) for any significant influence of the lakes on

the groundwaters. The justification for not considering the salt lakes as an important source of dSi is now provided in the revised MS in LNs 269-274.

**RC12. Fig. 6: It has been used to show older groundwaters contain high δ30Si due to more preferential removal of lighter isotopes on Fe-Al oxyhydroxides. However, the groundwater 25-90 has the lowest δ30Si (1.72) and much lower 3H (0.04) among older deep water. This misleading figure contradicts Fig. 3b indicating in situ release of lithogenic Si (low δ30Si).**

We thank the reviewer for highlighting this and we have modified our interpretation accordingly .

We agree that lithogenic silica is released in the deeper aquifers. We have also revised the presentation of the mixing model, which is now presented in Figure 4B in the revised MS.

**RC13. Section 5.4: The additional source of Si (high δ30Si) from the salt lakes excluded in this study can potentially bias dissolved Si isotope composition for the marine SGD component.**

We outlined our justifications for why we consider that the groundwaters are not connected to the salt lakes in AR11, namely the lake sediments/algal mats acting as aquitards and the lack of geochemical evidence for lakewater input in groundwaters. Regarding the marine SGD component, the salt lakes likely represent a low concentration source of dSi and, therefore, we do not think that this input would bias dissolved Si isotope composition for the marine SGD component. This is stated in the revised MS in LNs 269-274.

**RC14. Table A2. Replace this table with Box plots of saturation indicies calculated for all samples and include Fe-Mn-Al minerals.**

At the request of the reviewer, the saturation indices were recalculated using the water4f.dat database in PHREEQC and a box plot of these is provided in Figure 7 of the revised MS. Indeed, the results show that the groundwaters are saturated in many key Fe oxide minerals, such as goethite and  $Fe(OH)_3$ .

**Suggestions and editorial corrections:**

**RC15.  Do not club multiple parenthesis.**

The instance of this we could find was a mistake on LNs 37-38 in the original MS. This is now amended in the revised MS.

**RC16. Conservative mixing is misleading due to in situ Si release and thus change it to theoretical mixing.**

"Conservative mixing" is now amended to "theoretical mixing" in the revised MS.

***RC17. Line 118: delete space after "stabilized"***

This erroneous space has been deleted in the revised MS.

***RC18. Line 131: add cm after "18.2 megaOhm"***

This has been updated in the revised MS to '18.2 MΩcm'.

***RC19. Line 147: add Si after 28 and 29***

Si has been added after 28 and 29 so in the revised MS it now reads: '$^{28}$Si, $^{29}$Si and $^{30}$Si'.

***RC20. Line 285: Delete "and northwestern"***

This has been deleted in the revised MS.

***RC21. Line 327 and elsewhere: change "rock-water" to "aquifer-water"***

"Rock-water" has been updated to "aquifer-water" in the modified manuscript where applicable, except in the introduction where we also discuss riverine samples and "aquifer-rock" would not be appropriate.

**REVIEWER #2**

***Review of manuscript HESS-2020-429 submitted to Hydrology and Earth System Sciences by Martin and colleagues: The evolution of stable silicon isotopes in a coastal carbonate aquifer, Rottnest Island, Western Australia***

***Martin et al present silicon isotope ratios (expressed as δ30Si) from fresh and saline groundwater samples from a carbonate island in Western Australia. They interpret variation in δ30Si of the freshwater samples to be driven by sorption of Si to Fe/Al oxides, while the saline water samples are governed by mixing with a low [Si], high d30si endmember. Given a recent focus on topics like e.g. boundary exchange, coastal filtering, SGD, etc. for the trace element and isotope budgets of the ocean, this paper has potential to be an interesting case study.***

***In general, the manuscript is well written and seems appropriately referenced. The topic is an interesting one, I think, though does seem to fall slightly outside the scope of HESS as I understand it. I have some methodological concerns (detailed below), and some comments about the interpretations, some more major than others. I list these below in order they appear in the manuscript.***

***Comments***

***RC22. L71: this suggestion of interaction with basement rocks is interesting but never returned to. If it can be relevant for Li, why not for Si?***

The reviewer makes an excellent point that we had overlooked and may explain some of the lower $\delta^{30}Si$ values of the T2 groundwaters, relative to what would be expected for theoretical mixing of 5-90 (highest dSi groundwater) with seawater. This is now discussed in the revised MS in Section 5.3.2 in LNs 358-362.

***RC23. L95: relevance?***

In L95, we described how the recent decrease in rainfall and previous sea-level high stand events caused seawater intrusion into aquifer. The relevance of this was to highlight that intruded seawater may have adsorbed cations onto the aquifer matrix, which may be released into solution via cation exchange processes. This is now explicitly stated in the revised MS in LNs 100-102.

***RC24. L100, 103: spell out units here – TU = tritium units? pMC = percent modern carbon? G PES = ??***

The units are now spelled out for TU and pMC. 'G PES' was a typing error and should have read 'g $L^{-1}$'. These errors are now corrected in the revised manuscript.

***RC25. L104 vs. L98 – repetition, but with age expectations reversed. Not sure what's going on here.***

This was a mistake whereby the second instance on L104 should have read "T2 groundwaters" rather than "fresh groundwaters". This is now amended in the revised MS.

***RC26. L127: Some indication of precision and accuracy needed here.***

The accuracy of cations and anions measurements were evaluated the charge balance error, with 80% of the samples falling within ± 5% and all samples falling within ± 6.2%, which is much greater than the uncertainty than the analytical precision of measurements. This is now included in the revised MS (LN133-135).

*RC27. L132: This column protocol works sufficiently well for samples with low anionic components in the matrix. But for the saline samples, anions like sulphate or chlorine would elute at the same time as the sample. This could cause matrix effects between brack- eting NBS28 standards and samples. Previous work has shown that matrix effects, sensu lato, can induce large bias to silicon isotope ratios (e.g. Hughes et al. 2011 JAAS. What steps were taken to correct for this, or demonstrate that it is not a problem? Typically, a pre-concentration step like the so-called 'MAGIC' protocol is used for brackish and marine samples, since this also has the effect of removing much of the matrix.*

We acknowledge that this is a very important point raised by the reviewer that we had not addressed. This is now addressed in the Appendix of the revised MS in Section A.1.1 and discussed below.

We assessed the potential for anionic matrix effects in our groundwater analyses by assessing of there are correlations between groundwater $\delta 30Si$ values, sample SO4/Si molar ratios, and measured signal intensity for 29Si (before dilution to match bracketing standards) during MC-ICPMS measurements. As van den Boorn et al. (2009, JAAS) showed that the measured intensities of the Si isotopes varied positively with $SO_4/Si$ and $\delta^{30}Si$, these are useful criteria for assessing the potential anionic matrix effects. We found no statistically significant correlations (p-value <0.05) between these parameters and these figures are now presented in the Appendix of the revised manuscript (Fig. A.2). Thus, we found no evidence for an offset in groundwater $\delta^{30}Si$ values due to anionic matrix effects. This is consistent with results from Georg et al. (2006) who found that doping with 50 ppm $SO_4^{2-}$ ($SO_4^{2-}/Si$ molar ratio of ~14.4) did not induce any Si isotopic fractionation, and perhaps suggests that a brucite co-precipitation method would only be required for contents with much higher TDS contents, e.g., seawater and brines, whereas the meteoric groundwaters analysed in this study were at least ~30-fold lower than for local seawater, which had a molar $SO_4/Si$ ratio of ~9,000. Grasse et al. (2017, JAAS) also discuss some of these problems and present an inter-lab comparison of Si isotopes in seawater silicic acid, although we stress that all our groundwaters were meteoric and less saline than seawater. In any case, Grasse et al state that "Georg et al. and de Souza et al. did not observe a significant matrix effect of $SO_4^{2-}$ on their MC-ICP-MS analyses of Si isotopes from freshwater and seawater samples". Thus, anionic effects may not always a problem during Si isotopic analyses and Hughes et al. concluded that "our results confirm that the extent of matrix effect cannot be directly transferable from one laboratory to another, probably because of a combination of different instrument and sample types, analytical settings, and/or purification processing".

***RC28. L137: Drying to 'incipient' dryness is also slightly worrying. What does this mean, in practice? If silica precipitates because it becomes oversaturated in the solution, there is a danger that it will not redissolve in 2% HNO3. And because the precipitation is likely associated with a fractionation, this may also induce bias. Did the authors demonstrate 100% Si yield at this stage of protocol?***

To obtain sufficiently high concentrations for Si isotopic measurements, and not overload the capacity of the resin by loading a high volume of sample, we gently evaporated the solutions at 80 degC until a small wet blob of solution remained. Extreme care was taken not to completely evaporate the solution and form precipitates. No precipitates were observed for the redissolved samples and our procedure was completely HF-free. Our column yields following this method were confirmed to be 100±5% (analytical error). Standards were also processed in this manner.

***RC29. L140 or around: Give details of procedural blanks, for column chemistry and for alkali fusion***

As the limit of detection for Si at our facilities was relatively high (3.6 µM), we could not quantify the Si concentrations directly. Instead, we compared the signal intensities of our total procedural blanks for column chemistry (including blanks from alkali fusion that were processed through column chemistry) with our acid blanks by MC-ICPMS as a screening process and did not detect any measurable Si signal.

We now state in the revised MS that the $^{30}$Si signal intensities for total procedural blanks could not be distinguished for the background measured in 2% (v/v) HNO$_3$, typically <130 mV compared to standard and sample intensities of ~25 V. Thus, the blank contribution was less than 0.3% and all Si loaded onto the columns was eluted in the Si fraction.

***RC30. L147: Give 'Si' for 28 and 29.***

These are now stated as '$^{28}$Si, $^{29}$Si' in the revised manuscript.

***RC31. L148: M/ΔM = 2000 is probably at the low end of resolution with which the polyatomic interferences can be avoided, and could induce some noise to the measurement if some of the interference peaks overlap onto the measurement plateau (though the three-isotope plot shows this isn't a large problem – uncertainties should be given in this plot).***

As stated by the reviewer, we believe that the plot of $^{30}$Si/$^{28}$Si vs $^{29}$Si/$^{28}$Si supports the accuracy of our measurements. The uncertainties for measurements are now provided in the revised MS in Figure A.1.

***RC32. L155: This implies only 1 bracket was measured – is this correct? Normally, 3-5 individual standard-sample brackets are analysed per sample. If this is correct, the stated ±0.12 uncertainty may be too precise.***

As already stated in our response to RC2, it would have been preferable to measure samples in triplicate, but unfortunately resources allocated for analyses in the project were limited by budget and time constraints. Following the suggestion of reviewers 1 and 2, we now include the internal uncertainty (2 standard error; 2 s.e.) of the $\delta^{30}$Si measurement for each sample in Table 1 of the revised MS and this is provided above in the response to RC2.

***RC33. L158: There are more recent values for IRMM-018 – see e.g. Geilert et al. 2020 (Nat Comms); Baronas et al. 2018 (EPSL), etc.***

We thank the reviewer for bringing this to our attention. The $\delta^{30}$Si values for IRMM-018a from Geilert et al., (2020) and Baronas et al. (2018) were $-1.46 \pm 0.09$‰ and $1.57 \pm 0.08$‰, which are similar to our measured values. This is now stated in the revised manuscript.

***RC34. L161: Not sure the uncertainties of ±0.000 are required here.***

These uncertainties have been deleted in the revised MS.

***RC35. Figure 2: Panel A and B – the colorbar scale seems to imply the concentrations are negative.***

We understand the confusion due to the distance between the tick marks and the numbers. This has been increased in the revised figure in the revised MS.

***RC36. Figure 2: It's not clear which data are included in this. There is a red line in Fig 1, but presumably the samples falling off this line have been incorporated somehow. More detail could be useful.***

We apologise for the lack of clarity regarding which data was included in the plot. In the revised MS, we now state that all Rottnest Island groundwaters that we analysed were included to create these plots. To show the plots in two dimensions, we reduced sample locations onto a central axis (same longtitude) by interpolation and then created a grid based on their longitude and depth. This is now clearly stated in the Figure 2 caption in the revised MS.

***RC37. L172: I find the conclusion that T1 DSi is higher than fresh and T2 very hard to believe given Fig. 3A.***

We have double checked our calculations and got the same results. We also wanted to be sure originally and hence wanted to confirm that the difference was statistically valid, which is

confirmed by the p values (0.001 and 0.002) being below our significance threshold of 0.05. On Figure 3A, you can also see there are two T1 groundwaters with much higher dSi concentrations than the fresh groundwaters and also the majority of the T1 groundwaters plot above the T2 groundwaters, excluding 5-90.

***RC38. I am also skeptical about the correlation on L171.***

Again, we have double checked our calculations to confirm our result. This p value of 0.04 is clearly close to our defined significance threshold of 0.05, but nevertheless below it, and should be regarded as statistically valid in this context.

***RC39. L191: extra ")"***

We thank the reviewer for highlighting this typing error and it is now corrected in the revised MS.

***RC40. Results section: I can't see the Si concentration and δ30Si for the seawater sample mentioned on L114. In e.g. Fig 4B, a literature value is used from Singh et al., but this is not sufficiently justified.***

The data for the seawater sample is now included in Table 1 in the revised MS. Unfortunately, due to the very low dSi concentration of local seawater at Rottnest Island, the $\delta^{30}$Si was not measured as a preconcentration procedure to remove the seawater matrix, such as MAGIC, was not established at our facilities.

The value for $\delta^{30}$Si from Singh et al. (2015) was included as it was the only available data from the Indian Ocean. Based on the well-established, inverse relationship between dSi and $\delta^{30}$Si (Grasse et al., 2013; Singh et al., 2015), we now use the average value of +1.7‰ for low-concentration seawater sample from GEOTRACES (Grasse et al., 2017).

***RC41. L262: I disagree that there is a threshold of Al concentration necessary for Si isotopes to fractionate – pure SiO2 experiments also show fractionation (as noted elsewhere in this manuscript).***

We agree that we may have focussed too much on the results from Oelze et al. that only found Si isotopic fractionation in the presence of Al and not properly considered findings from previous studies, e.g., Geilert et al. (2014; GCA). We have now removed this statement in the revised manuscript.

***RC42. L264: If the interpretation for d7Li is the same as for δ30Si, why is there no relationship between the two? (L189)***

We thank the reviewer for pointing out the inconsistency in our reasoning. In the revised MS, we now highlight the lack of correlation between $\delta^7Li$ and $\delta^{30}Si$ in RI groundwaters, and present the following hypothesis to explain this, namely the contrast between high concentrations of Li in seawater relative to fresh meteoric groundwaters, whereas dSi is a nutrient and depleted in seawater. This is an important distinction on RI since modern seawater intrusion (Bryan et al., 2016), and past sea-level high stands (~2 m higher than present), e.g., events at ~4 and 7 ka (Coshell and Rosen, 1994; and Gouramanis et al., 2012), would have probably intruded seawater into the shallow groundwater system. Such seawater intrusion episodes would be expected to adsorb Li, but not Si, onto the aquifer matrix during previous/ongoing and provide Li to groundwaters through cation exchange processes.

***RC43. L265: the evidence for Fe oxides playing an important role seems to be circumstantial. Is there any direct evidence they are present, with sufficient sorption capacity, to be important?***

As outlined in our response to RC14, we have recalculated our saturation indices using phreeqc. This shows show that the groundwaters are saturated in many key Fe oxide minerals, such as goethite and $Fe(OH)_3$. A box plot for all minerals is included in the revised MS and included in the discussion (Figure 7).

***RC44. Section 5.2: There is no real explanation for the positive trend between Si concentration and isotope ratios – this is a bit counter-intuitive. One might expect a negative correlation between Si concentration and isotope ratio.***

Reviewer 1 in RC6 presented an interesting point about the potential role of diffusion in modulating the groundwater $\delta^{30}Si$ values of fresh on Rottnest Island. We have attempted to further clarify this point in the revised MS. We had already attempted to discuss the role of physical mixing and diffusion in Section 5.2 in LNs 284-297. We have attempted to further clarify this point in the revised MS.

***RC45. L305: missing word (divided?)***

We thank the reviewer for pointing out this mistake and have corrected it in the revised MS.

***RC46. L312: See also above – this does not appear to be the case from Fig 3A.***

The reviewer here is referring to our statement that the T1 groundwaters have significantly higher concentrations than the fresh and T2 groundwaters. As we outlined in RC38, and stated in the

original MS, this statement is supported by statistical tests that show the p values are less than our threshold for statistical significance (0.05).

**RC47. Section 5.3.1. I struggle to follow the rationale for including 'T1' groundwaters as a separate case rather than just viewing them as intermediate between freshwaters and the marine/T2 samples.**

We agree with the reviewer and have removed "T1 groundwaters" from the title of Section 5.3.1 in the revised MS.

**RC48. L335 and around – the challenge seems to be to define the non-marine endmember. If this was more rigorously achieved, then the discussion might be more usefully structured in terms of deviations from conservative mixing as in e.g. estuary papers (Zhang et al. 2020 GCA).**

We thank the reviewer for bringing this paper to our attention. We agree that having a well-defined fresh/non-marine end-member would be advantageous. However, it can be seen from Fig 3A and 3B in the original MS that dSi concentrations and Si isotopic ratios do not show a simple mixing relationship with Cl concentrations (and by extension, salinity). Moreover, the dSi concentrations and Si isotopic ratios of the fresh groundwaters were highly variable, despite have very low salinities. This might reflect the fact that RI has a small carbonate island aquifer, which is not comparable to a large river, such as the Amazon or Yangtze, flowing into an ocean or sea.

**RC49. In general, it's also not clear where the marine endmember comes from. I don't think a value from the Bay of Bengal is necessarily very representative for waters bathing western Australia (see e.g. Holzer et al. 2015 GBC)**

As stated in our response to RC40, we now use the average value of $+1.7‰$ for low-concentration seawater sample from GEOTRACES (Grasse et al., 2017).

**RC50. L355: Could/should this not instead yield low $\delta 30Si$ values if there is a re-equilibration between solute and rock?**

We agree with the reviewer and now state in the revised MS that low $\delta^{30}Si$ in the deeper aquifer may result from the equilibration processes between the groundwaters and the rock in Section 5.3 of the revised MS.

**RC51. L372: I don't understand this – wouldn't these values be representative of the terrestrial component of SGD?**

We agree with the reviewer and this is now considered to represent the terrestrial SGD component , i.e., 'fresh SGD'.

**RC52. It is disappointing there is no attempt to calculate fluxes and place them in a global context. In general, the global implications section is rather weak and seems added on as an afterthought. It would be good to see a better discussion of how the lessons from this case study can (or cannot) be applied elsewhere, for example, and a more rigorous attempt at upscaling.**

We agree with the reviewer and now estimate this in the revised manuscript. The terrestrial SGD flux from Western Australia is estimated to be 3.2 km³/a with carbonate aquifers comprising 50% of the coastline (Zekster et al. 2007; Rahman et al., 2019). By adopting the average dSi concentration of the fresh groundwaters (112±23 µM), we estimate that the dSi flux from the Tamala Limestone is 1.79 x10$^8$ ± 0.37 x10$^8$ mol/a of dSi, corresponding to ~27% of the dSi flux from Western Australia.

**REVIEWER #3**

**RC53. The manuscript is well written, providing a critical dataset in an understudied system. I enjoyed reading it. The figures are generally clear. I have suggested some minor revisions and have some suggestions for the discussion. There were some few grammatical errors. I don't anticipate the revisions would take the authors very long to incorporate.**

We thank the reviewer for their kind comments and have addressed their individual comments on a point-by-point basis in the following section.

**RC54. Suggested changes throughout the paper: I prefer "bSi" to "BSi" as the capital "B" in "BSi" can be mistaken for the symbol for Boron. Similarly, I suggest changing "DSi" to "dSi" so that the capital D won't be mistaken for Deuterium. For readers versed in the biogenic silica literature, the capital letters may not pose a problem. However, for new readers it may contribute to some confusion.**

We had not considered that this may be misleading, but agree with the reviewer's suggestion. This is now changed in both this document and the revised MS.

**RC55. Well written introduction. Clear reasoning, concise review, and compelling set-up and motivation for the project to fill a known data gap. I suggest adding Ehlert et al, 2016 (Ehlert, C., Reckhardt, A., Greskowiak, J., Liguori, B.T., Böning, P., Paffrath, R., Brumsack, H.J. and Pahnke, K., 2016. Transformation of silicon in a sandy beach ecosystem: Insights from stable silicon isotopes from fresh and saline groundwaters. Chemical Geology, 440, pp.207-218) to the 2nd paragraph (lines 42-54). That data was not included in Frings et al 2016.**

We thank the reviewer for highlighting that this data was not included in Frings et al. 2016 and agree that it is an important reference. We have now included this citation in the introduction.

**RC56. 2. Study Area The last paragraph is confusing w.r.t. description of "fresh" mixing type. One set of parameters is given for this zone in lines 99-101, and another set is given later in the paragraph (lines 103-105), with some repeated conditions (e.g., above 1m AHD).**

This was a mistake whereby the second instance on L104 should have read "T2 groundwaters" rather than "fresh groundwaters". This was already addressed in response to RC25 and is now correctly written in the revised MS.

**RC57. Also would it be possible to add a salinity or [Cl-] range to the three mixing zones?**

We thank the reviewer for their suggestion. The Cl concentrations have been added in lines 105 and 109 in the revised MS; they are also now provided in Table A.1.

**RC58. 3. Methods: No salt corrections or formation of a brucite precipitate was conducted to account for matric effects prior to purification on via Biorad AG 50WX8?**

We only processed samples that had sufficiently high dSi and low TDS so that they could be processed without exceeding the resin capacity, typically <10%, and a brucite coprecipitation was not necessary. A further discussion of potential artefacts from anions during Si isotopic measurements is also provided in Section A.1.1.

**RC59. 4.Results Do the authors have a surface coastal water/open ocean value for dSi? I think the manuscript would be improved by adding a comparison of d30Si found in fresh-T2 to Ehlert et al 2016 in the second paragraph (lines 176-186).**

The data for the seawater sample is now included in Table 1 in the revised MS. Unfortunately, due to the very low dSi concentration of local seawater at Rottnest Island, the $\delta^{30}$Si was not measured as a preconcentration procedure to remove the seawater matrix, such as MAGIC, was not established at our facilities.

As suggested, we have also now included a comparison to the data from Ehlert et al. in the revised MS in lines 207-209.

**RC60. Incidentally, have the authors seen Mayfield et al., 2021 (Nature Communications)?**

The authors were aware of this paper as we actually provided some samples from Rottnest Island that form part of the dataset, but this was published after we submitted. It is now appropriately cited in the revised MS in Lines 46-47.

**RC61. Line 190: missing punctuation at end of sentence Line 191: extraneous parenthetical after "respectively"**

We thank the reviewer for pointing this out and this is now corrected in the revised manuscript.

***RC62. Fig. 2: So cool! Regarding the legend for TDS and dSi: are those negative values?***

We thank the reviewer for their kind words and understand the confusion due to the distance between the tick marks and the numbers. This distance has been increased in the revised figure for clarity.

***RC63. Fig. 4: Can authors add a regression line to 4a as this is discussed in results and discussion?***

A regression line has been added to Figure 4a in the revised MS.

***RC64. Also, in 4b, there are 2 green dashed lines and only description of 1 green dashed line in the caption.***

We thank the reviewer for noticing this and have updated the caption for Figure 4b in the revised MS accordingly.

***RC65. 5.1 Evidence for the source of DSi. . . Lines 223-226: referring to a figure here would be helpful***

This is a good suggestion and a reference to Figure 4a has been provided in the revised MS.

***RC66. Would you expect lower d30Si if RI was behaving like a closed system which could be described via Rayleigh distillation model? To clarify, if almost all the dSi supplied by quartz dissolution formed a secondary clay mineral, would the subsequent dissolution of that secondary mineral would impart a dissolved d30Si signature more like the primary mineral? This scenario could be added to the discussion of secondary mineral dissolution in lines 230-235), unless the authors have other evidence this is not likely.***

The reviewer raises an interesting and important point. However, in the revised MS, as our recalculated phreeqc saturation indices show that groundwaters are indeed saturated in clay minerals, it might be expected that once formed they would not be subsequently dissolved. For this reason, we do not focus heavily on this topic in the revised MS.

***RC67. Line 237: regarding bSi solubility, coastal settings can actually make for recalcitrant diatom frustules: proximity to dissolved Al and metals can help preserve frustules (e.g., solubility studies by Van Cappellen, Van Bennekom, and Loucaides et al 2012). I suggest authors clarify this sentence with specific conditions of their study site or leave out the last phrase about preservation in coastal sediments.***

In light of the reviewer sharing their expertise and what has been found in the relevant published works, our statement on the preservation of diatoms in coastal sediments has been removed.

*RC68. Could the authors add some site specific data w.r.t. to dissolved Fe concentrations in their discussion of adsorption being the likely mechanism that takes up light isotope of dSi.*

As outlined in our response to RC4, we now include these Fe data in the revised MS in Table 1. The revised version of Table 1 is also provided in our response to RC2.

*RC69.  What were the concentrations of Fe in the 0.5M HCl leachate?*

The data from the leaching experiments is provided below (Martin et al., 2020). As readers may not have access to this publication, we will provide this in the supplementary information of the revised MS.

| | Sample | Li | Sr | Ca | Al/Ca | Fe/Ca | K/Ca | Li/Ca | Mg/Ca | Na/Ca | Si/Ca | Sr/Ca | Zr/Ca | Sr/Li |
|---|---|---|---|---|---|---|---|---|---|---|---|---|---|---|
| | | uM | uM | mM | uM/m | uM/m | uM/m | uM/m | uM/m | uM/m | uM/m | uM/m | uM/m | uM/u |
| B01 | Acid soluble | 0.01 | 0.54 | 0.22 | 0.46 | 0.23 | 1.89 | 0.07 | 71.90 | 0.20 | 0.11 | 2.45 | 0.10 | 36.03 |
| S01 | Acid soluble | 0.01 | 0.55 | 0.22 | 0.62 | 0.15 | 0.29 | 0.06 | 79.27 | 0.20 | 0.05 | 2.48 | 0.05 | 41.70 |
| B01 | Residue | 0.52 | 16.35 | 3.90 | 5.01 | 5.19 | 2.58 | 0.13 | 115.70 | 1.44 | 4.95 | 4.20 | 2.90 | 31.40 |
| S01 | Residue | 0.33 | 10.13 | 2.78 | 12.84 | 3.11 | 9.04 | 0.12 | 124.01 | 2.02 | 10.75 | 3.65 | 7.64 | 30.28 |
| B01 | Bulk | 0.48 | 12.77 | 3.38 | 3.44 | 3.69 | 2.80 | 0.14 | 109.15 | 1.66 | 0.68 | 3.78 | 6.28 | 26.42 |
| S01 | Bulk | 0.35 | 13.61 | 3.54 | 5.10 | 1.29 | 4.03 | 0.10 | 122.92 | 1.58 | 3.37 | 3.84 | 3.99 | 38.52 |
| Av. | Acid soluble | 0.01 | 0.54 | 0.22 | 0.54 | 0.19 | 1.09 | 0.06 | 75.58 | 0.20 | 0.08 | 2.47 | 0.07 | 38.8 |
| Av. | Residue | 0.42 | 13.19 | 3.46 | 4.27 | 2.49 | 3.42 | 0.12 | 116.03 | 1.62 | 2.03 | 3.81 | 5.14 | 32.47 |
| Av. | Bulk | 0.18 | 7.08 | 1.88 | 2.86 | 0.72 | 2.16 | 0.08 | 101.1 | 0.89 | 1.71 | 3.16 | 2.02 | 40.11 |

[a]1SD uncertainty for all elements is ±6.2%.

*RC70.  Are there dissolved Fe or O2 profiles at the sites? Are these sediments oxic?*

As outlined in our response to RC4, we now include these Fe and DO data in the revised MS in Table 1. The revised version of Table 1 is also provided in our response to RC2. These data show that the shallow groundwaters are generally oxic but become sub-oxic in the deeper T2 groundwaters.

*RC71.  The more positive d30Si of dSi observed in groundwater in Ehlert et al 2016 was due to secondary mineral formation. Can the authors provide more evidence of why that's not occurring here? PHREEQC doesn't encompass the solubilities of all the amorphous-type alumino-silicates that can form.*

As outlined in our responses to RC14 and RC43, we have recalculated our saturation indices using phreeqc. This shows show that the groundwaters are saturated in many key Fe oxide minerals, such as goethite and  Fe(OH)$_3$. The saturation indices were recalculated to include all Fe-Mn-Al minerals. As the reviewer highlights, many common Fe-Al-Mn phases (such as

Ferrihydrite, Lepidocrocite, etc.) are not included in the phreeqc database that we originally used (phreeqc.dat). Thus, we have remodelled all saturation states using water4f.dat, which includes many of these minerals. Indeed, the results show that the groundwaters are saturated in many key Fe oxide minerals, such as goethite and $Fe(OH)_3$. These are provided in Figure 7 of the revised MS.

***RC72. Line 286: refer to a figure Line 305: missing a word or phrase here ". . .can be into two water types. . ."***

We thank the reviewer for pointing out this mistake. "Divided" has been added after "can be" in the revised MS.

***RC73. Line 325-328: This three end-member mixing concept is important: could it be highlighted in one of the figures and accompanied by a reference to that figure at the end of this sentence? Or is this in Figure 4b?***

We did attempt to show this to some degree in Figure 4B. In the revised manuscript, a three end-member mixing plot has been added to Figure 4B.

***RC74. Conclusions highlight broad impacts, well written.***

We thank the reviewer for their kind feedback.

**References**

Bryan, E., Meredith, K.T., Baker, A., Andersen, M.S., Post, V.E. and Treble, P.C. (2020) How water isotopes (18O, 2H, 3H) within an island freshwater lens respond to changes in rainfall. Water research 170, 115301.

Grasse, P., Brzezinski, M.A., Cardinal, D., De Souza, G.F., Andersson, P., Closset, I., Cao, Z., Dai, M., Ehlert, C. and Estrade, N. (2017) GEOTRACES inter-calibration of the stable silicon isotope composition of dissolved silicic acid in seawater. Journal of Analytical Atomic Spectrometry 32, 562-578.

Grasse, P., Ehlert, C. and Frank, M. (2013) The influence of water mass mixing on the dissolved Si isotope composition in the Eastern Equatorial Pacific. Earth and Planetary Science Letters 380, 60-71.

Martin, A.N., Meredith, K., Norman, M.D., Bryan, E. and Baker, A. (2020) Lithium and strontium isotope dynamics in a carbonate island aquifer, Rottnest Island, Western Australia. Science of The Total Environment 715, 136906.

Playford, P.E. (1997) Geology and hydrogeology of Rottnest island, Western Australia.

Singh, S.P., Singh, S.K., Bhushan, R. and Rai, V.K. (2015) Dissolved silicon and its isotopes in the water column of the Bay of Bengal: Internal cycling versus lateral transport. Geochimica et Cosmochimica Acta 151, 172-191.